# Early antiretroviral therapy favors post-treatment SIV control associated with the expansion of enhanced memory CD8+ T-cells

Caroline Passaes [1,2] ✉, Delphine Desjardins [3], Anaïs Chapel[1,2], Valérie Monceaux [1,2], Julien Lemaitre[3], Adeline Mélard [4], Federico Perdomo-Celis [2], Cyril Planchais [5], Maël Gourvès [1], Nastasia Dimant [3], Annie David[2], Nathalie Dereuddre-Bosquet [3], Aurélie Barrail-Tran [3,6], Hélène Gouget[3], Céline Guillaume[3], Francis Relouzat [3], Olivier Lambotte[3,7], Jérémie Guedj [8], Michaela Müller-Trutwin[2], Hugo Mouquet [5], Christine Rouzioux[9], Véronique Avettand-Fenoël [4,10], Roger Le Grand [3,11] & Asier Sáez-Cirión [1,2,11] ✉

HIV remission can be achieved in some people, called post-treatment HIV controllers, after antiretroviral treatment discontinuation. Treatment initiation close to the time of infection was suggested to favor post-treatment control, but the circumstances and mechanisms leading to this outcome remain unclear. Here we evaluate the impact of early (week 4) vs. late (week 24 post-infection) treatment initiation in SIVmac251-infected male *cynomolgus macaques* receiving 2 years of therapy before analytical treatment interruption. We show that early treatment strongly promotes post-treatment control, which is not related to a lower frequency of infected cells at treatment interruption. Rather, early treatment favors the development of long-term memory CD8+ T cells with enhanced proliferative and SIV suppressive capacity that are able to mediate a robust secondary-like response upon viral rebound. Our model allows us to formally demonstrate a link between treatment initiation during primary infection and the promotion of post-treatment control and provides results that may guide the development of new immunotherapies for HIV remission.

Despite its unquestionable success, antiretroviral therapy (ART) cannot completely eliminate HIV reservoirs, which remains a major barrier to HIV eradication. As a consequence, a rapid viral rebound is observed in most persons living with HIV (PLWH) when ART is interrupted[1–3]. Nevertheless, in a very limited number of individuals designated post-treatment HIV controllers (PTCs), viremia remains suppressed for prolonged periods after ART withdrawal[1,4–15].

Identifying immunological or virological signatures that could predict the time to viral rebound after stopping ART and understanding the mechanisms leading to post-treatment control is a priority to guide the development of innovative strategies for sustained HIV remission. Several markers, such as low cell-associated HIV DNA levels, high CD4/CD8 ratios, loss of functional plasmacytoid dendritic cells or low expression of immune checkpoint molecules, have been associated with delayed viral rebound after ART discontinuation[4,13,16–19], and distinctive transcriptomic and metabolomic signatures have been observed in post-treatment controllers[20,21]. However, the mechanisms underlying durable HIV control after ART interruption remain elusive.

The VISCONTI study provided the first comprehensive description of the virological and immunological features of PTCs and suggested that the immunological mechanisms driving post-treatment control might differ from those observed in natural HIV controllers[13]. The VISCONTI study indicated that ART initiation during the first weeks following infection and then maintenance for several years might favor post-treatment HIV control. Indeed, ART was initiated in a higher frequency of PTCs during primary HIV infection (PHI)[5,7,8,10], although ART was initiated in some PTCs during the chronic phase[4,15]. The CHAMP study confirmed a higher frequency of post-treatment control among individuals who were treated during early infection compared to those treated during chronic infection[11]. Early ART initiation limits the seeding of viral reservoirs and viral evolution, diminishes residual inflammation, protects against tissue damage and preserves immune responses[22–26]. However, viremia control upon treatment discontinuation is observed in only a fraction of PLWH with early ART administration, and ART initiation during the earliest stages of infection does not impede viral rebound after treatment interruption[27]. It remains unclear whether there is a window of opportunity for ART initiation that may limit the extent of seeding of HIV reservoirs without restraining the maturation of the developing HIV-specific immune response that may be later important for post-treatment control. The retrospective observational nature of cohort studies, the relatively low number of individuals with long-term post-treatment control status, an important level of heterogeneity regarding the time to ART initiation and ART regimens, the duration of antiretroviral treatment prior to treatment interruption, and the criteria for defining PTCs among the different cohorts have hampered the evaluation of the role of early ART and the factors that lead to post-treatment control[28].

Nonhuman primate (NHP) models of SIV infection closely recapitulate events occurring after HIV infection in humans, including the dynamics of viral reservoir seeding, development of immune responses and evolution upon antiretroviral treatment initiation[29–33]. NHP studies have provided valuable insights into the factors associated with HIV pathogenesis and control and potential strategies aimed at delaying viral rebound after analytical treatment interruption (ATI)[26,34–37]. NHP models offer a unique opportunity to study in homogeneous experimental conditions the impact of early initiation of multiyear ART, which was documented among the PTCs of the VISCONTI study[13]. We therefore explored here (pVISCONTI study) how early (week 4 post-infection, roughly the median delay to ART initiation that we had observed in the VISCONTI study) vs. late (week 24 post-infection) initiation of ART impacted the outcome after treatment discontinuation in *cynomolgus macaques* (CyMs, *Macaca fascicularis*) infected with SIVmac$_{251}$, and we performed extensive characterization of the virological and immunological features before and after ART. We showed that early ART initiation strongly favored durable post-treatment SIV control in this model, which was associated with mobilization of memory CD8$^+$ T cells with enhanced antiviral capacities against the rebounding virus.

## Results

### Early ART initiation favored delayed viral rebound and a high frequency of post-treatment SIV controllers

To assess the impact of early versus late ART initiation on viral control after treatment interruption, we monitored the kinetics of SIVmac$_{251}$ replication in CyMs in which ART was initiated either at 4 weeks p.i. ($n = 2 \times 6$) or at 24 weeks p.i. ($n = 2 \times 6$). After 24 months of ART, treatment was interrupted, and animals were monitored over 24 to 48 weeks (Fig. S7). As expected, all animals exhibited a primary SIV infection with a high plasma viral load (pVL) that peaked at 10 days p.i. (6.8 [6.5–7.0] log SIV-RNA copies/mL) (Fig. 1A). No differences in peak viremia were observed between the animals in the early- and late-treatment groups ($p = 0.11$) (Fig. 1A, C). Similar early viral dynamics

were observed in CyMs ($n = 17$) infected under the same experimental conditions and that never received ART (Fig. S7C). In the W4 group, ART was initiated during the declining phase of initial viremia, while in animals in the W24 group, pVL was already stable at the time of ART initiation (Fig. 1A, Fig. S7). Nevertheless, pVL did not differ overall between early- (5.1 [4.5–5.4] log SIV-RNA copies/mL) and late- (4.4 [3.7–5.7] log SIV-RNA copies/mL) treated macaques ($p = 0.4$) at the time of ART initiation (Fig. 1C). One animal (CB296A) from the early-treated group that was euthanized before ART following severe acute infection and one animal (BA979I) from the late-treated group with controlled viremia below 400 copies/mL before ART initiation (Fig. S7) were excluded from further analyses. The initiation of ART efficiently suppressed viremia, and the time to achieve undetectable pVL levels (defined as at least 3 consecutive pVL measurements <10 SIV RNA copies/ml) was comparable between groups (112 days [48-167] for W4-treated CyMs; 64 days [28-169] for W24-treated CyMs, $p = 0.47$). In one of the animals treated at W24 (BB425F), the pVL remained low but detectable despite ART (<100 SIV RNA copies/ml of plasma). pVL was undetectable during ART in all the other animals with some sporadic low-level blips (Fig. S7). The pVL at the time of ATI was similar between early- and late-treated animals ($p = 1$) (Fig. 1C). No differences were observed between the W4- and W24-treated macaques in the levels of cell-associated SIV RNA in blood or peripheral lymph nodes (PLNs) at this time (Fig. S8).

Following treatment interruption, significant differences in pVL dynamics were observed between the W4- and W24-treated groups. Overall, viral rebound with a rapid increase in pVL, reaching levels higher than 1000 copies/mL, was observed in all animals, except for two early-treated CyMs exhibiting maintenance of low viremia (CCB065 and BB9I) (Fig. 1B, D). The time to rebound (defined as the first pVL measurement higher than 400 copies/mL) was delayed by two weeks in the W4-treated animals (median time to viral rebound 28 days post ATI) when compared with the W24-treated group (median 14 days post-ATI) (Fig. 1B, D, G) ($p = 0.0003$). Moreover, significant differences in the magnitude of pVL were observed between the early- and late-treatment groups at all time points following viral rebound post-ATI (Fig. 1D). After ART interruption, late-treated CyMs exhibited a higher peak of pVL (all reaching pVL > 10,000 copies/mL) ($p = 0.0066$) (Fig. 1E) and were exposed to higher levels of cumulative viremia ($p = 0.004$) (Fig. 1F) than early-treated CyMs. The levels of pVL progressively declined in most early-treated macaques but remained relatively stable and high for most late-treated animals (Fig. 1D). Before the end of the study, 9 out of 11 (82%) W4-treated animals exhibited controlled pVL levels below 400 SIV-RNA copies/mL, indicating the status of post-treatment controllers (PTC), whereas this was the case for only 2/11 (18%) W24-treated animals (Fig. 1G) ($p = 0.008$). All PTCs had at least two pVL measurements of <400 copies RNA/ml after viral rebound control, and all, except BA736J (last pVL=514 RNA/ml), exhibited pVL maintenance of <400 copies until the end of the study. Of note, among the 17 untreated CyMs that were studied in parallel, only 2 (12%) naturally exhibited a pVL of <400 copies/ml, while the 15 others stably maintained high-level viremia (>10$^4$ copies/ml) (Fig. 1G, Fig. S7). Although animals carrying the M6 MHC haplotype, strongly favoring natural SIV control in this model (e.g.[36]), were not included in the pVISCONTI study, we cannot exclude that other immunogenetic factors may have influenced the outcome of treatment interruption. However, the early- and late-treated groups of this study were matched based on their MHC haplotypes, and therefore, the overall differences in the outcomes of the groups were predominantly related to the delay in treatment initiation. Our results showed that 2 years of ART initiated at W4 p.i. strongly enhanced the probability of post-treatment SIV control when compared to ART initiated at W24 or to the rates of spontaneous SIV control in this model.

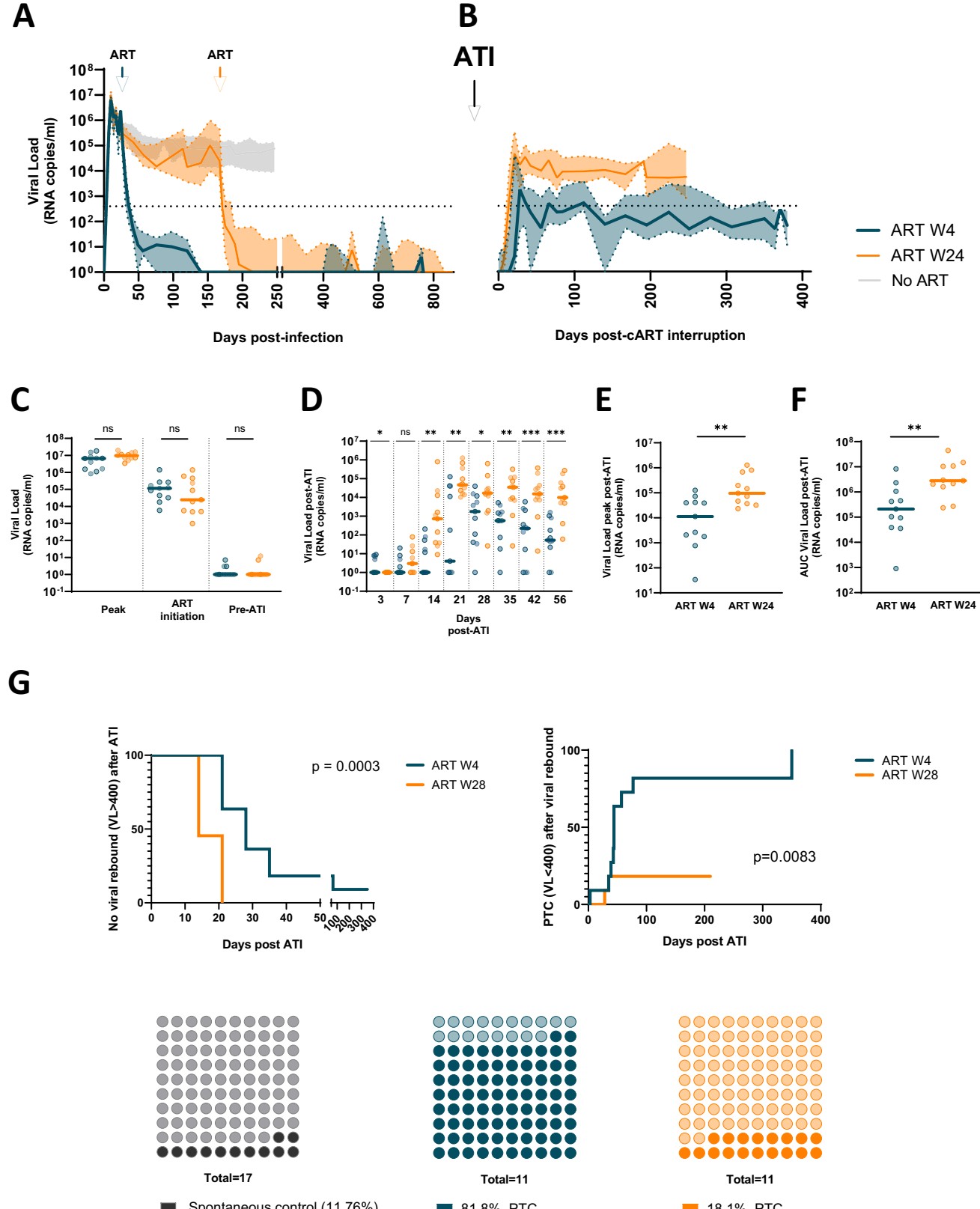

### Early treatment leads to lower SIV reservoir levels and preservation of CD4⁺ T cells after treatment interruption

We explored whether the time of ART initiation influenced CD4⁺ T-cell counts or cell-associated SIV-DNA levels before ART interruption, which could explain differences in viral rebound.

CD4⁺ T-cell-associated SIV-DNA levels closely mirrored the dynamics of plasma viremia. High levels of CD4+ T-cell-associated SIV-DNA were observed during primary SIV infection in the blood of W4- and W24-treated CyMs (Fig. 2A). As expected, SIV-DNA levels in CD4⁺ T cells were high at treatment initiation in the W4 macaques since

**Fig. 1 | Early therapy is associated with a higher rate of post-treatment controllers.** Plasma viral load kinetics **A** prior to ART initiation and during ART; and **B** post-ATI in W4-treated (*n* = 11) and W24-treated CyMs (*n* = 11). Plasma viral load kinetics during the initial six months following infection in animals not receiving ART (*n* = 17) shown as reference. Medians and IQR are shown. **C** Comparison of plasma viral load levels between W4- and W24-treated CyMs at the peak (acute infection), at the time of ART initiation and prior to treatment interruption. **D** Comparison of plasma viral load levels between W4- and W24-treated CyMs in the early days following ATI. The magnitude of the plasma viral load post-ATI in W4- and W24-treated CyMs is indicated by (**E**) the viral load peak and by (**F**) the cumulative pVL post-ATI (area under the curve - AUC, considering all pVL

measurements until 6 months post-ATI). **C–F** Values for individual animals (W4 *n* = 11, W24 *n* = 11) and medians are shown. *\*p* < 0.05, *\*\*p* < 0.01; *\*\*\*p* < 0.001; ns non-significant; 2-sided Mann–Whitney U test. **G** Kaplan–Meier analyses of maintaining no viral rebound (time without pVL > 400 copies/ml) following ART interruption (left) and achievement of post-treatment control (time to durable pVL <400 copies/ml after viral rebound) (right) in W4- and W24-treated macaques (*n* = 11 for each). Frequency of CyMs spontaneously controlling plasma viremia (<400 copies/mL) in untreated SIV infection and after antiretroviral treatment interruption (middle and right panels). The frequencies of posttreatment controllers among W4- and W24-treated CyMs are shown. Mantel Cox log-rank test was used to compare the two groups. Source data are provided as a Source Data file.

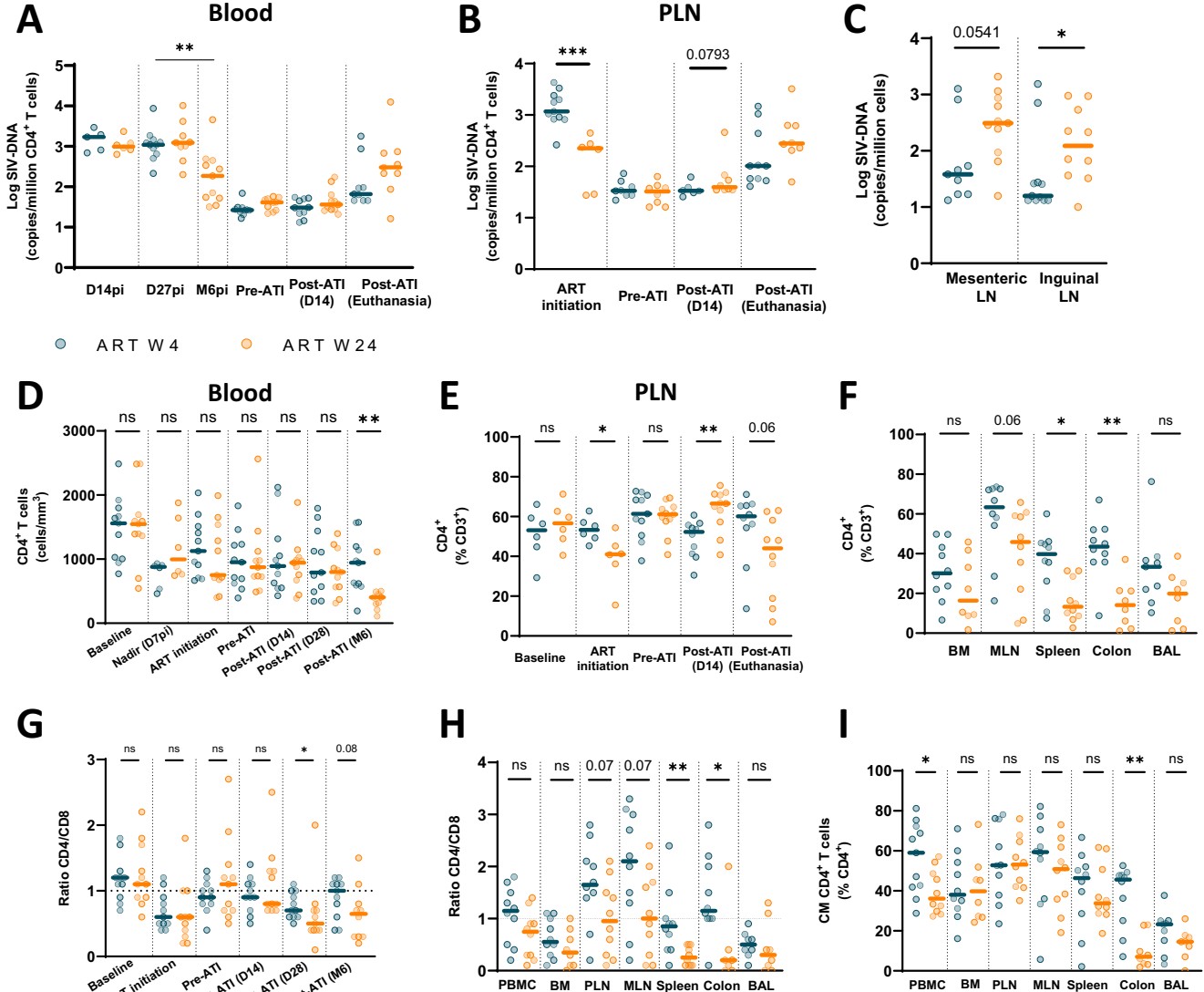

**Fig. 2 | Post-treatment controllers maintain low number of infected cells and high CD4+ T cell frequencies.** Kinetics of SIV-DNA levels in **A** blood and in **B** PLNs in W4-treated and W24-treated CyMs. The results are expressed as copies of SIV-DNA/million CD4+ T cells. **C** Levels of SIV-DNA in mesenteric and inguinal lymph nodes at euthanasia. The results are expressed as copies SIV-DNA/million cells. Longitudinal evolution of CD4+ T cells in (**D**) blood and in (**E**) PLNs in W4- and W24-treated CyMs. The results are shown as absolute CD4+ T-cell counts in blood and as the proportion of CD4+ T cells among CD3+ lymphocytes in PLNs. **F** Proportion of CD4+ T cells among CD3+ lymphocytes in BM, MLN, spleen, colon mucosa and BAL

at euthanasia. Longitudinal evolution of the CD4/CD8 ratio in (**G**) blood. **H** The ratio of CD4/CD8 in PBMCs, BM, PLNs, MLNs, spleen, colon mucosa and BAL at euthanasia. The dashed line indicates ratio = 1. **I** Proportion of central memory (CD45RA⁻CD27⁺CCR7⁺) CD4+ T cells in PBMCs, BM, PLNs, MLNs, spleen, colon mucosa and BAL at euthanasia. **A–I** Individual values (*n* = 6 or 11 for each group depending on sampling during the pVISCONTI-1 and pVISCONTI-2 study phase). Medians are shown. *\*p* < 0.05, *\*\*p* < 0.01; *\*\*\*p* < 0.001; ns non-significant; Two-sided Mann–Whitney U test. Source data are provided as a Source Data file.

these animals were still in the transition phase between primary and chronic infection. Consequently, lower levels were found at the time of ART initiation in the blood and peripheral lymph nodes (PLNs) of W24 macaques (Fig. 2A, B). However, no difference was observed in SIV-DNA levels between early- and late-treated macaques at the time of ART interruption (Fig. 2A, B). Following ART discontinuation, an increase in SIV-DNA levels was observed, particularly in W24-treated animals, which, at the end of the study, tended to have higher levels of SIV-DNA in blood, mesenteric and inguinal lymph node cells than W4-treated animals (Fig. 2A–C). To take into account the potential influence of the animals with a divergent outcome within the W4- and W24-treated groups, we compared the PTC and non-PTC animals after ART discontinuation irrespective of the time of ART initiation (Fig. S9A–D). No significant differences were observed in the levels of CD4+ T-cell-associated SIV-DNA between PTCs or non-PTCs during primary infection or before ART interruption in blood or lymph nodes (Fig. S9E, F). SIV-DNA levels strongly increased in non-PTCs after ART interruption, reaching levels similar to those found before ART initiation, while the levels in PTCs remained close to those found during ART (Fig. S9E–G).

Similarly, no differences were found in the dynamics of CD4+ T-cell counts at primary SIV infection between animals in the W4- and W24-treated groups (Fig. 2D, E). In blood, the nadir CD4+ T-cell counts were observed at 7 days p.i. A trend toward lower CD4+ T-cell counts in blood and a lower proportion of CD4+ T cells in PLNs were observed in the late-treated group at the time of ART initiation. However, no difference was observed between the groups at the time of ATI in blood and in PLNs (Fig. 2D, E). Following ATI, CD4+ T-cell counts did not decrease within the first weeks; rather, a gradual decline was observed in late-treated animals, resulting in significant differences between groups at week 24 post-ATI in blood (Fig. 2D), in PLNs (Fig. 2E) and in other lymphoid and nonlymphoid tissues analyzed at the time of euthanasia (BM, MLN, spleen, colonic mucosa, and BAL) (Fig. 2F). Overall, higher CD4+ T-cell levels in blood and tissues were maintained in PTCs at the end of the study, irrespective of the time of treatment initiation. This was related neither to differences occurring during primary infection nor at the time of ART interruption in PTC versus non-PTC macaques (Fig. S9H–J).

We further characterized the impact of early vs. late ART initiation on the T-cell compartment by analyzing the CD4/CD8 ratio and the frequency of central memory (CM) CD4+ T cells in blood and tissues, two parameters that have been suggested to be associated with post-treatment control in human cohorts and NHPs in previous studies[13,17,38]. As expected, an inversion of the CD4/CD8 ratio was observed upon SIV infection in all animals (Fig. 2G). The ratio was normalized after two years on ART independent of the time of ART initiation, and no differences were found between the W4- and W24-treated groups at the time of ATI. However, a significant decrease was noted in the W24-treated group upon ART discontinuation, resulting in a lower CD4/CD8 ratio in W24- vs. W4-treated animals at the end of the study in lymph nodes, spleen and colon (Fig. 2H). Likewise, higher frequencies of CM CD4+ T cells were detected in early-treated animals in blood and tissues at the end of the study (Fig. 2I).

Collectively, our results show that PTCs, which significantly predominated in macaques with ART initiation early during SIV infection (W4), were characterized by durable maintenance of low SIV-DNA levels and pananatomical preservation of the CD4+ T-cell compartment when compared to non-PTCs, which predominated in the group of macaques with ART initiation in established chronic infection (W24). These differences were absent at the time of ART interruption.

## Magnitude of the anti-gp140 antibody response after exposure to viremia

We next explored whether early ART initiation in CyMs may have influenced the maturation of adaptive immune responses developing during primary infection, which may have later facilitated immune control of viral rebound. Hence, we first determined the levels of anti-SIV Env antibodies over time in the plasma of W4- and W24-treated macaques (Fig. 3A). Weak anti-gp140 IgG titers were comparably detected in early- and late-treated macaques at week 4 p.i. (Fig. 3A, B). The production of anti-SIV Env IgGs was halted by ART initiation in the animals treated at this time, while the titers continued to increase in the W24-treated animals (Fig. 3A). ART initiation led to a drop in anti-gp140 IgG titers in all animals, but higher IgG titers were detected among the W24-treated animals prior to treatment interruption (p = 0.0032) (Fig. 3B). IgG titers sharply increased following viral rebound after ART interruption in all animals, except for CCB065 (W4-treated group), which did not experience strong viral relapse (Fig. 3A). IgG titers increased faster after ATI than following infection in both the W4- and W24-treated macaques (Fig. S10). The overall magnitude of the anti-SIV IgG response during the 24 weeks following ATI was significantly higher among the W24-treated macaques than among the W4-treated macaques (p = 0.0018) (Fig. 3C) and higher among non-PTCs than among PTCs irrespective of the time of ART initiation (Fig. S9K). Moreover, when all animals were considered together, the IgG titers post-ATI positively correlated with the magnitude of viral rebound (r = 0.6552; p = 0.0009−Fig. 3D).

We did not find significant anti-SIV gp140 IgA levels at week 4 p.i. in either early- or late-treated macaques (Fig. 3E). In contrast, anti-gp140 IgA antibodies were detected by week 24 p.i. in the late-treated animals (Fig. 3F). After two years of ART, anti-SIV Env IgA titers did not differ from baseline levels, and no differences were observed between W4- and W24-treated macaques prior to treatment interruption (Fig. 3F). Similar to the anti-gp140 IgG titers, IgA titers increased post-ATI faster than during primary infection (Fig. S10), but no difference in the global levels of anti-SIV Env IgA antibodies post-ATI was observed between W4- and W24-treated animals (Fig. 3G). As opposed to IgG titers, no correlation was found between the magnitude of anti-gp140 IgA titers post-ATI and the magnitude of viral rebound (Fig. 3H).

Altogether, these results indicate that SIV Env-specific memory B cells persisted during prolonged treatment and were reactivated by viral rebound following treatment interruption, favoring a prompt antibody response per se. Although anti-Env IgG and IgA antibodies were not sufficient to limit viral rebound, further functional analyses are warranted to evaluate the impact of ART initiation on the quality of SIV-specific IgA/IgG responses.

## Early treatment enhanced the capacity of CD8+ T cells to suppress SIV after ART interruption

We next longitudinally evaluated the functional activity of SIV-specific CD8+ T cells by analyzing the ex vivo capacity of CD8+ T cells to suppress SIV infection in autologous CD4+ T cells (Fig. 4A, Fig. S11)[39], an approach that has allowed us to reveal the presence of highly efficient CD8+ T cells in spontaneous HIV and SIV controllers[36,40–42]. The SIV suppressive activity of CD8+ T cells was weak in the blood (Fig. 4A, B) and in PLNs (Fig. 4E) during primary infection. At week 4 p.i., the CD8+ T-cell-mediated SIV-suppressive activity in blood did not differ from baseline in either the early- or late-treated groups. These results reflect a limited antiviral capacity for SIV-specific CD8+ T cells induced during primary infection, confirming our previous observations in PLWH[43] and CyMs infected with SIVmac251[36]. A modest but significant increase in CD8+ T-cell-mediated SIV-suppressive activity was observed over months in the W24-treated group (p = 0.02) (Fig. 4A, B). ART initiation did not immediately impact the capacity of CD8+ T cells to suppress SIV infection, but a tendency toward higher CD8+ T-cell-mediated SIV-suppressive activity was observed in early-treated macaques before treatment interruption (blood p = 0.056), and the difference was more marked in PLNs (p = 0.005) (Fig. 4B).

Following treatment interruption, a strong increase in CD8+ T-cell-mediated SIV suppressive activity was observed in the W4- and W24-treated macaque groups, either in the blood (Fig. 4A, C; Fig. S11) or

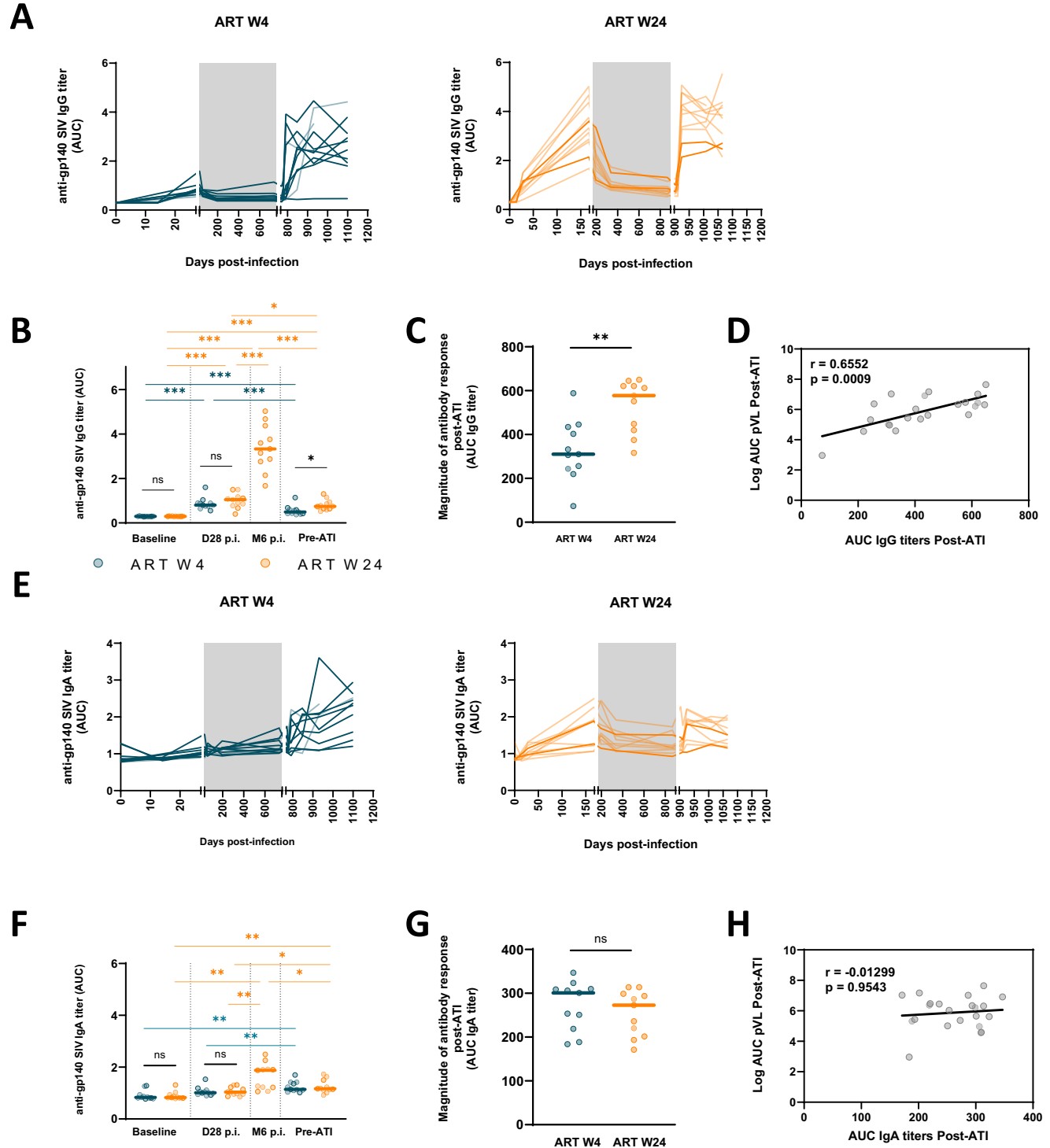

**Fig. 3 | The anti-SIV antibody levels after treatment interruption are associated with the magnitude of antigen stimulation.** Kinetics of plasma anti-gp140 SIV IgG (**A**) and IgA (**E**) in W4-treated and W24-treated CyMs. Comparison of anti-gp140 SIV IgG (**B**) and IgA (**F**) levels between W4- and W24-treated CyMs at baseline, on day 28 p.i.; at 6 months p.i. for the W24-treated group, and prior to ART interruption. **C** The magnitude of humoral response post-ATI is indicated by the cumulative plasma anti-gp140 SIV IgG (**C**) and IgA (**G**) measurements (area under the curve - AUC until 6 months post-ATI). Spearman correlation between pVL AUC post-ATI and anti-gp140 SIV IgG (**D**) and IgA (**H**) AUC post-ATI. **A**–**H** Individual values ($n = 11$ animals per group are shown. **B**, **C**, **F**, **G** Medians are shown, *$p < 0.05$, **$p < 0.01$; ***$p < 0.001$; ns non-significant; Two sided Mann–Whitney U test. Source data are provided as a Source Data file.

PLNs (Fig. 4E). The overall magnitude of the CD8+ T-cell-mediated SIV-suppressive activity during the 24 weeks that followed ATI tended to be higher in the early-treated group ($p = 0.09$) (Fig. 4D), and the maximum levels of SIV suppressive activity achieved by CD8+ T cells after ART interruption were higher during this period for the W4-treated vs. the W24-treated CyMs (Fig. 4C). The CD8+ T-cell-mediated SIV-suppressive capacity was also higher in PLNs of W4-treated macaques at the end of the study (Fig. 4E). A stronger capacity of CD8+ T cells to suppress SIV infection was observed in the blood, PLNs and spleen of PTCs when compared to non-PTCs irrespective of time

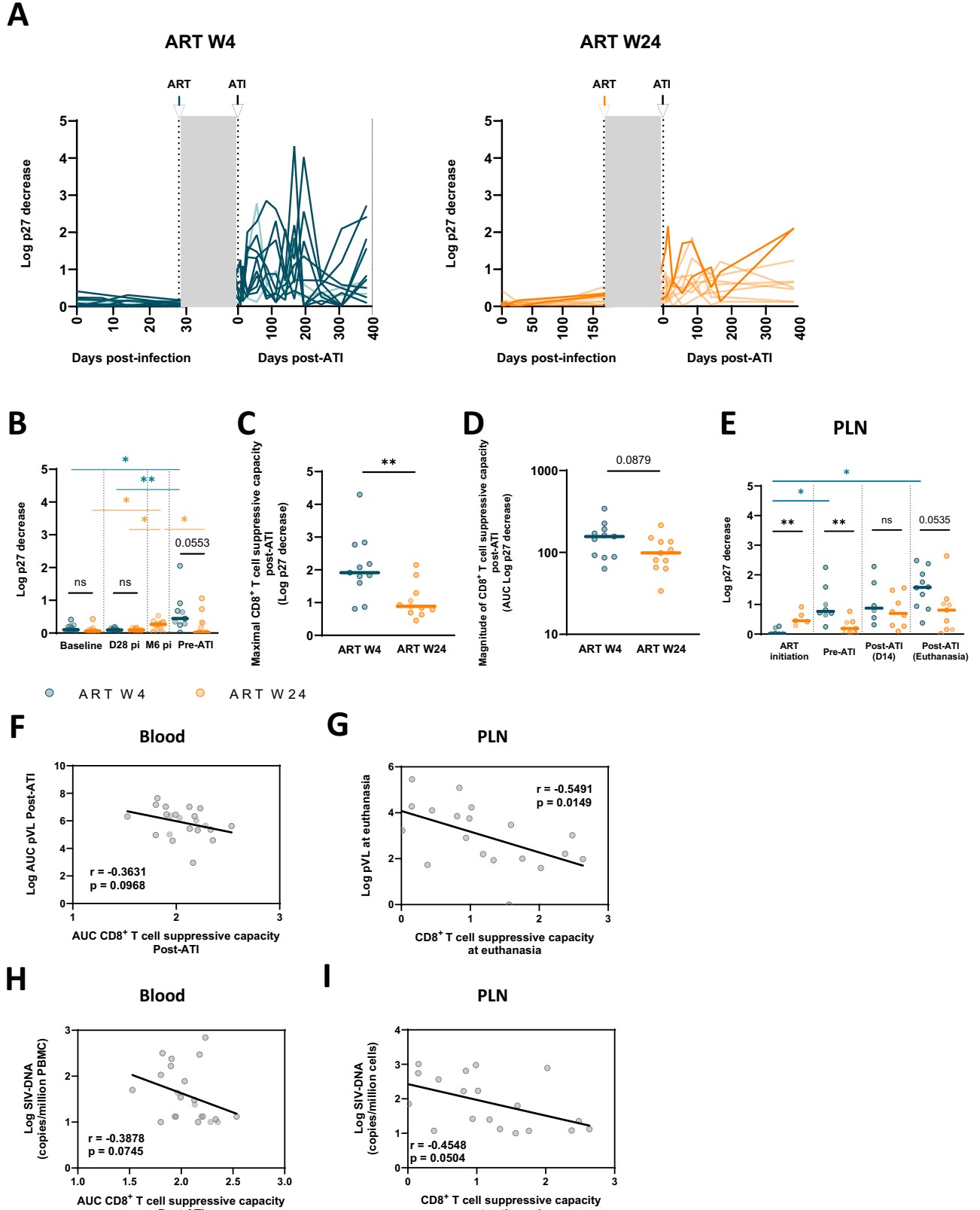

to ART initiation (Fig. S9L–N). Finally, when all animals were considered together, the overall magnitude of the CD8[+] T-cell-mediated SIV-suppressive activity post-ATI in blood tended to negatively correlate with the magnitude of viral rebound (r = −0.4; p = 0.097 – Fig. 4F) or the frequency of CD4 carrying SIV-DNA (r = −0.39; p = 0.075 – Fig. 4H). A negative correlation was observed between the capacity of

CD8[+] T cells of PLNs to suppress SIV infection at the end of the study and the pVL (r = −0.55; p = 0.015 – Fig. 4G) or the frequency of infected cells (r = −0.56; p = 0.016 – Fig. 4I) at this timepoint.

Our results indicate that prolonged ART favored the enhancement of the antiviral potential of CD8[+] T cells, which were able to exert stronger activities upon viral rebound than during primary infection.

**Fig. 4 | CD8+ T cells mobilized after interruption of early treatment have enhanced SIV suppressive capacity. A** Kinetics of CD8⁺ T-cell-mediated SIV-suppressive activity in the blood of W4-treated and W24-treated CyMs at primary SIV infection and post-ATI. **B** Comparison of CD8⁺ T-cell-mediated SIV-suppressive activity between W4- and W24-treated CyMs at baseline, on day 28 p.i.; at 6 months p.i. for the W24-treated group, and prior to ART interruption. **C** Comparison of the maximal CD8⁺ T-cell-mediated SIV-suppressive activity during the period following treatment interruption in W4- and W24-treated CyMs. **D** The magnitude of CD8⁺ T-cell-mediated SIV-suppressive activity is indicated by the cumulative measurements post-ATI (area under the curve – AUC of all measurements until 6 months post-ATI). **E** Kinetics of CD8⁺ T-cell-mediated SIV-suppressive activity in PLNs of W4- and W24-treated CyMs at the time of ART initiation, 14 days post-ATI and at

euthanasia. The results are shown as log p27 decrease in the presence of CD8⁺ T cells. Spearman correlation between (**F**) pVL AUC post-ATI and blood CD8⁺ T-cell-mediated SIV-suppressive activity AUC post-ATI, **G** pVL and PLN CD8⁺ T-cell-mediated SIV-suppressive activity at euthanasia, (**H**) SIV-DNA levels in blood CD4 at euthanasia and blood CD8⁺ T-cell-mediated SIV-suppressive activity AUC post-ATI, **I** SIV-DNA levels at euthanasia and PLN CD8⁺ T-cell-mediated SIV-suppressive activity at euthanasia. (**A–I**) Individual values (*n* = 11 animals per group, except for (E), where *n* = 5–10 per group were analyzed depending on sample availability) are shown. (**B–E**) Medians are shown; *\*p* < 0.05, \*\**p* < 0.01; \*\*\**p* < 0.001; ns non-significant; Two sided Mann–Whitney U test. Source data are provided as a Source Data file.

Our results indicate that this effect was stronger in PTC macaques, in which CD8⁺ T cells were able to produce superior SIV suppressive activity in blood and lymphoid tissues and were favored by early ART initiation. These results suggest a role for CD8⁺ T cells in limiting viral rebound and the number of infected cells and a potential association with the establishment of post-treatment SIV control.

### Interruption of early-initiated ART was followed by the expansion of weakly activated memory CD8⁺ T cells

To better understand the striking differences in the antiviral capacity of CD8⁺ T cells observed during primary infection and those observed in the same animals after ART interruption, we compared the phenotypic characteristics of CD8⁺ T cells during these periods. We first evaluated cell activation as the proportion of CD8⁺ T cells coexpressing CD38 and HLA-DR. In agreement with previous reports in human cohorts and NHP models[30,36,44,45], we found very strong CD8⁺ T-cell activation during primary SIV infection in all animals (Fig. 5A). The frequency of activated cells remained high and relatively stable in the late-treated macaques, but ART initiation rapidly reduced CD8⁺ T-cell activation in both W4- and W24-treated animals. Upon ART interruption, a rapid increase in CD8⁺ T-cell activation was observed in the W24-treated group, reaching levels similar to those observed during primary infection (Fig. 5A, B). In contrast, no or little CD8⁺ T-cell activation was observed for the animals in the W4-treated group (Fig. 5A). Lower overall CD8⁺ T-cell activation levels were observed between primary infection (AUC weeks 0-4 p.i.) and post-ATI for early-treated macaques (AUC weeks 0–4 post ATI), whereas no differences were noticed for the late-treated group (Fig. 5B). A trend toward higher CD8⁺ T-cell activation levels was observed in PLNs of W24-treated animals at the end of the study (*p* = 0.12) (Fig. 5C). No correlation was found between CD8⁺ T-cell activation levels and their capacity to suppress SIV infection post-ATI (Fig. 5D).

High frequencies of proliferating (Ki67⁺) CD8⁺ T cells were also observed during primary SIV infection (Fig. 5E). The frequency of these cells decreased with the transition to chronic infection in the animals treated at week 24 p.i., then stayed stable until ART initiation and was drastically reduced following ART initiation in all cases. Ki-67 expression was lower post-ATI than during primary infection, both in W4- and W24-treated animals (Fig. 5E, F), but overall higher levels of proliferating CD8⁺ T cells were found early post-ATI in the W4-treated macaques than in the W24-treated macaques in blood (*p* = 0.004) (Fig. 5F) and PLNs (*p* = 0.009) (Fig. 5G). Of note, a positive correlation (*r* = 0.76, *p* = 0.006) was found between the rate of proliferating CD8⁺ T cells after ART interruption and their capacity to suppress SIV infection during the same period (Fig. 5H).

We then investigated differences between primary infection and post-ATI in the CD8⁺ T-cell subpopulations. Primary infection was characterized by the predominant expansion of CD8⁺ T cells with an effector memory (EM) or effector phenotype (Fig. 6A, B). In contrast, CD8⁺ T cells with a central memory (CM) phenotype were expanded after treatment interruption in W4-treated macaques, while the proportion of EM and effector CD8⁺ T cells remained low (Fig. 6A). A

higher proportion of CM CD8⁺ T cells was also found after ART interruption than during primary infection in PLNs of early-treated macaques (Fig. 6C). For the animals treated at week 24 p.i., an expansion of EM and effector CD8⁺ T cells was again observed in blood and PLNs after ART interruption (Fig. 6B, D). No differences were observed in the frequency of CM CD8 + T cells between W4- and W24-treated macaques at the time of ART interruption (*p* = 0.8, Fig. 6A, B), but CM CD8+ T cells were more frequently found in the blood and PLNs of W4-treated macaques 14 days after ATI (*p* = 0.013 and p = 0.002, for CD8 + T cells with the CM phenotype in W4- vs. W24-treated macaques in blood and PLNs, respectively). Moreover, higher levels of proliferating cells were found among CM CD8⁺ T cells in the blood and PLNs of W4-treated macaques after ART interruption than in W24-treated macaques (Fig. 6E). A positive correlation was found between the proportion of CM CD8⁺ T cells post-ATI in blood (day 28 post-ATI) and PNLs (day 14 post-ATI) and the capacity of CD8⁺ T cells to suppress SIV infection at the same time points (*r* = 0.58, *p* = 0.02; *r* = 0.72, *p* = 0.01 – Fig. S12) or the cumulative SIV suppressive capacity post-treatment interruption (*r* = 0.49, *p* = 0.05 in blood; *r* = 0.78, *p* = 0.004 in PNL – Fig. 6F). No correlation was found between the capacity of CD8⁺ T cells to suppress SIV infection and the proportion of naïve, transitional memory (TM) or EM CD8⁺ T cells (Fig. S12), but a trend toward a negative correlation was found with the proportion of effector CD8⁺ T cells (r = −0.43, *p* = 0.086).

Therefore, while strongly activated effector CD8⁺ T cells with limited antiviral potential were expanded during primary infection, early and prolonged ART appeared to promote the preservation or development of CD8⁺ T cells with central memory characteristics that were able to rapidly expand after ART interruption and to exert potent antiviral activities with limited activation.

### SIV-specific CD8⁺ T cells with enhanced survival and proliferative capacities were mobilized after ART interruption in early-treated macaques

We next explored the characteristics of CD8⁺ T cells that specifically reacted to stimulation with pools of SIV peptides in vitro (Fig. S3). SIV-specific CD8⁺ T cells producing IFNγ, TNFα or IL-2 or expressing CD107 in response to 6 h of SIV peptide stimulation were detected after primary SIV infection in all animals analyzed (Fig. 7A). The frequency of SIV-specific CD8⁺ T cells decreased as expected during ART but progressively increased after viral rebound, and no differences in the magnitude or the polyfunctionality of these cells were observed between W4- and W24-treated animals before ART initiation or after ART interruption (Fig. 7A, B). No difference was observed either in the magnitude or polyfunctionality of SIV-specific CD8⁺ T cells during primary infection vs. after ART interruption that could explain the striking increase in the SIV-suppressive activity of CD8⁺ T cells observed for all macaques after ATI (Fig. 7A–C). In contrast, we found higher expression levels of the IL-7 receptor (CD127) on SIV-specific CD8⁺ T cells from W4-treated macaques than on W24-treated macaques after ART interruption, suggesting the presence of long-living memory cells in the former (Fig. 7C). Similarly, no difference was

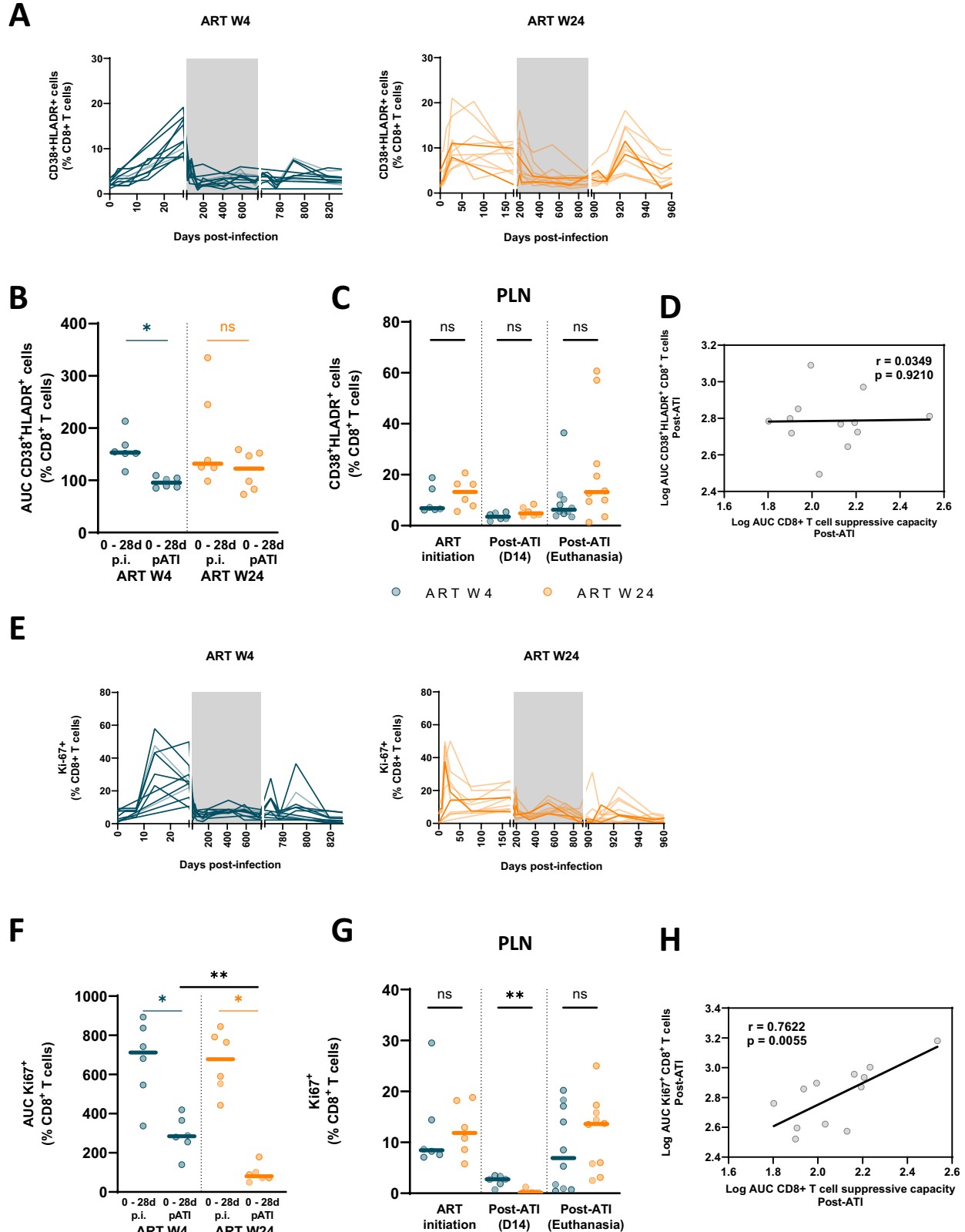

**Fig. 5 | CD8+ T cells expanding after interruption of early treatment express low levels of activation markers.** Expression levels of CD38 and HLA-DR activation markers (**A**) and intracellular levels of Ki67 proliferation markers (**E**) in blood CD8+ T cells during primary SIV infection and post-ATI in W4-treated (n = 11) and W24-treated CyMs (n = 11). Cumulative CD38, HLA-DR (**B**) and Ki67 (**F**) expression levels in blood CD8+ T cells during the first 4 weeks following SIV infection and post-ATI. Comparison of CD38 and HLA-DR (**C**) and Ki67 (**G**) expression levels in PLN CD8+ T cells between W4- and W24-treated CyMs at the time of cART initiation, 14 days post-ATI and at euthanasia. The results are shown as percent frequencies of CD8+ T cells or AUC; Spearman correlation between cumulative (AUC) CD38 and HLA-DR (**D**) and Ki67 (**H**) expression levels post-ATI and blood CD8+ T-cell-mediated SIV-suppressive activity AUC post-ATI. (A-H) Individual data are shown (n = 6 animals per group); (**B**–**D**, **F**, **G**) Medians are shown; *p < 0.05; **p < 0.01; ***p < 0.001; ns non-significant; Two sided Mann–Whitney U test. Source data are provided as a Source Data file.

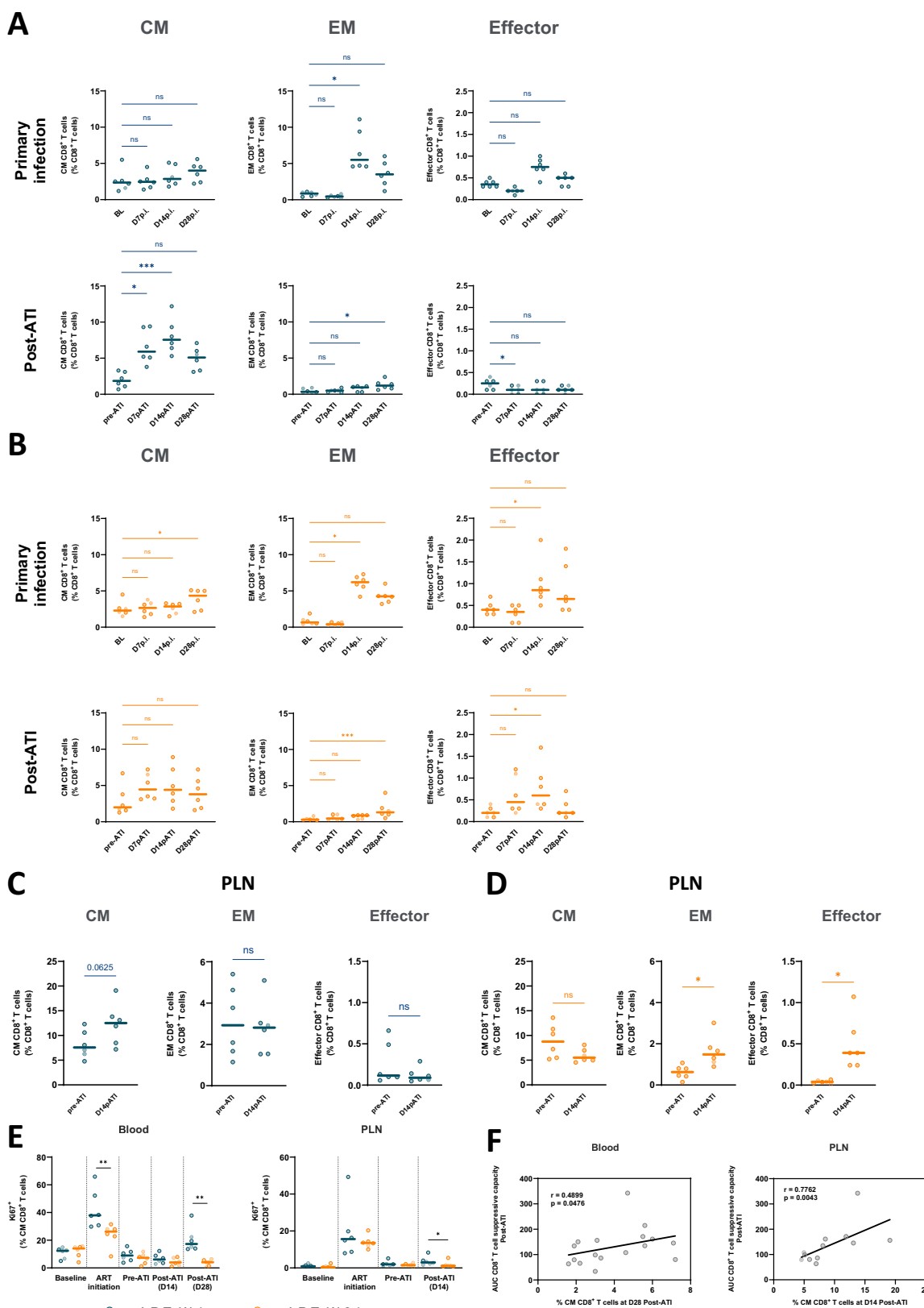

observed in the magnitude of the SIV-specific CD8+ T-cell response between PTCs and non-PTCs; however, SIV-specific CD8+ T cells from the former expressed higher levels of CD127 at the end of the study (Fig. S9O, P).

Further analysis of the memory phenotype of the SIV-specific CD8+ T cells at this time (Fig. S4) showed that memory (CD45RAneg)

SIV-specific CD8+ T cells from the W4-treated macaques were distinguished by higher expression of CCR7 and lower expression of CD39 than cells from W24-treated macaques. We also found a trend toward higher expression of the transcription factor TCF-1, which has been associated with stemness potential, in the cells from the W4-treated macaques (Fig. 7D). A higher expression of CCR7 and TCF1 on

**Fig. 6 | Expansion of central memory CD8 + T cells is observed in blood and lymph nodes after interruption of early treatment.** Evolution of central memory (CM) (CD45RA⁻CD27⁺CCR7⁺), effector memory (EM) (CD45RA⁻CD27⁻CCR7⁻) and effector (CD45RA⁺CD27CCR7⁻) CD8⁺ T-cell subsets in blood at primary SIV infection (top) and early post-ATI (bottom) in W4- (**A**) and W24-treated (**B**) CyMs. Frequency of CM, EM and effector CD8⁺ T cells in PLNs of W4- **C** and W24-treated **D** CyMs prior to and 14 days post-ATI. **E** Comparison of intracellular levels of Ki67 in blood (left) and PLN (right) CM CD8⁺ T cells at baseline, at the time of ART initiation, prior to and post-ATI in W4- and W24-treated CyMs. **F** Spearman correlation between blood CD8⁺ T-cell-mediated SIV-suppressive activity AUC post-ATI and the frequency of CM CD8⁺ T cells 28 days post-ATI in blood (left) and 14 days post-ATI in PLNs (right). **A–F** Individual data are shown (n = 6 animals per group); **A–E** Medians are shown; *p < 0.05, **p < 0.01; ***p < 0.001; ns non-significant; **A, B** Friedman test with Dunn's correction; C,D Two sided Wilcoxon matched-pairs rank test. **E** Two sided Mann–Whitney U test. Source data are provided as a Source Data file.

SIV-specific memory CD8+ T cells was even more evident in animals classified as PTCs vs. non-PTCs (Fig. S9Q). The cells from non-PTCs were, in contrast, characterized by higher levels of PD-1. These results corroborated the presence after ART interruption of a higher fraction of memory SIV-specific CD8+ T cells with long-lived and stem-like traits in the W4-treated animals, while a more differentiated, prone-to-exhaustion profile was present in the cells from the late-treated animals.

To more precisely evaluate the memory potential of SIV-specific CD8⁺ T cells, we sequentially stimulated circulating CD8⁺ T cells collected from early- and late-treated macaques with SIV peptides in vitro at different timepoints in our study and evaluated the proliferative capacity of responding cells in a 6-day-interval experiment (Fig. S5). We found that in general, SIV-specific CD8⁺ T cells present after ART interruption in W4-treated macaques showed a higher division and proliferation index than cells from W24-treated macaques (Fig. 7E). Moreover, we found that, for the early-treated macaques, the proliferative capacity of SIV-specific CD8⁺ T cells was stronger after ART interruption than during primary infection (Fig. 7E). We then used Phenograph unsupervised clustering analysis to explore the characteristics of these SIV-specific CD8⁺ T cells. Phenograph analyses identified 7 potential clusters of SIV-specific CD8⁺ T cells (Fig. 7F). For W4-treated macaques, the frequency of Cluster 1 cells was decreased, and those of Clusters 2 and 3 were increased post-ATI when compared to pre-ART (Fig. 7G). In contrast, no differences in cluster frequencies were observed over time for the W24-treated animals (Fig. S6). Cells in Clusters 1 and 3 shared relatively similar expression levels of the CD45RA, CCR7 and CD27 differentiation markers. However, Cluster 1, which was more represented during primary infection in W4-treated macaques, was characterized by higher expression of IFNγ and pS6 (a marker of activation of the mTORC1 pathway), while Cluster 3, which was more frequently present after ART interruption, was characterized by higher levels of CD127 and TNFα (Fig. 7H). Cells in Cluster 2 also differed from those in Cluster 1 in terms of high expression of TNFα and low expression of IFNγ, IL-2 or pS6 but expressed higher levels of CD45RA. All three clusters were characterized by the presence of cells with high levels of pAKT (a marker for the activation of the mTORC2 pathway) (Fig. 7H). Of note, we have recently associated the capacity to produce TNFα sustained by activation of mTORC2 without upregulation of mTORC1 as a hallmark of CD8⁺ T cells with stemness and memory potential that are frequently found in people with natural control of HIV infection[46,47].

Collectively, our results indicate that early treatment initiation in SIV-infected CyMs favored the establishment of memory-like SIV-specific CD8⁺ T cells with enhanced proliferative and survival capacity that could mediate an efficient secondary response to rebounding virus after ART interruption. These observations suggest an association between the memory potential of SIV-specific CD8⁺ T cells during ART and the development of post-treatment SIV control.

## Discussion

In this study, we show that 2 years of ART initiated at week 4 post-infection strongly promoted durable post-treatment SIV control in CyMs infected with SIVmac₂₅₁. This beneficial effect of ART was largely lost when treatment initiation was deferred to 6 months post-infection. This study was specifically designed to assess the impact of early vs. late ART initiation on post-treatment control, and the experimental conditions were informed by our previous observations based on cohorts of PLWH[13,23,48].

The underlying hypothesis of the study was that in the context of early ART introduction, obtaining an equilibrium between limiting the extent and diversity of the viral reservoir and allowing the maturation of immune responses might be critical to achieving viral control after treatment interruption. Previous studies have shown that initiation of treatment during the eclipse phase of infection in macaques (2–4 days p.i.) or hyperacute infection (Fiebig I–II) in PLWH drastically limits the extent of viral spread, but it does not preclude viral rebound after treatment interruption[26,27]. Indeed, an extremely early treatment (before peak viremia is reached) most likely prevents the development of antiviral responses. Therefore, treatment initiation at this point might merely halt the virus and delay primary infection. In contrast, we hypothesized that starting treatment a little later, once immune responses have been primed, might allow an optimal maturation of the responses, in a context of rapidly decreasing viremia, and their preservation, while still impacting the formation of the viral reservoir if treatment is maintained long enough. The delay to treatment initiation (not too soon, not too late) and the length of the treatment would therefore be critical to achieve this balance[49]. We selected 4 weeks post-infection for the initiation of early treatment because this period roughly corresponded to the median time to treatment initiation that we had estimated for the post-treatment controllers in the ANRS VISCONTI study[13]. We planned to treat animals for 24 months, as we had also reported that maintaining ART over several years was important to successfully reduce viral reservoirs and achieve immune reconstitution in people in whom treatment was initiated during primary infection[48], and we had observed a steady decrease in the frequency of cells carrying HIV-DNA over two years after initiation of early ART[23]. A recent study also reported a continuous decay in the frequency of intact SIV genomes during the first two years of ART in SIV-infected macaques[31].

Our results showed an unprecedented frequency of post-treatment controllers among the animals in which treatment was started at week 4 post-infection. The rate of post-treatment control in our model was higher than previously anticipated from estimations calculated retrospectively in early-treated PLWH[10,11,13,50]. However, the incidence of post-treatment control was much higher than the rate of spontaneous SIV control without ART observed in this model under the same experimental conditions. Moreover, the advantage of early ART initiation was evidenced by the striking difference observed in the probability of post-treatment control between groups of macaques differing only by the delay to treatment initiation. The replication of the comparison between W4- and W24-treated animals in two different experimental phases with similar results makes our observation highly robust. Our study therefore formally establishes a role of early ART initiation in posttreatment control.

Although no predictive marker of post-treatment control has been validated thus far, some studies have suggested that various markers, such as lower viral loads pre-ART, lower HIV-DNA levels or higher CD4/CD8 ratios before ART interruption[17,19,38], might be associated with lower viremia or delayed rebound after ART discontinuation. As expected, all controls and early- and late-treated animals experienced similar primary infection, with equivalent viral load peaks

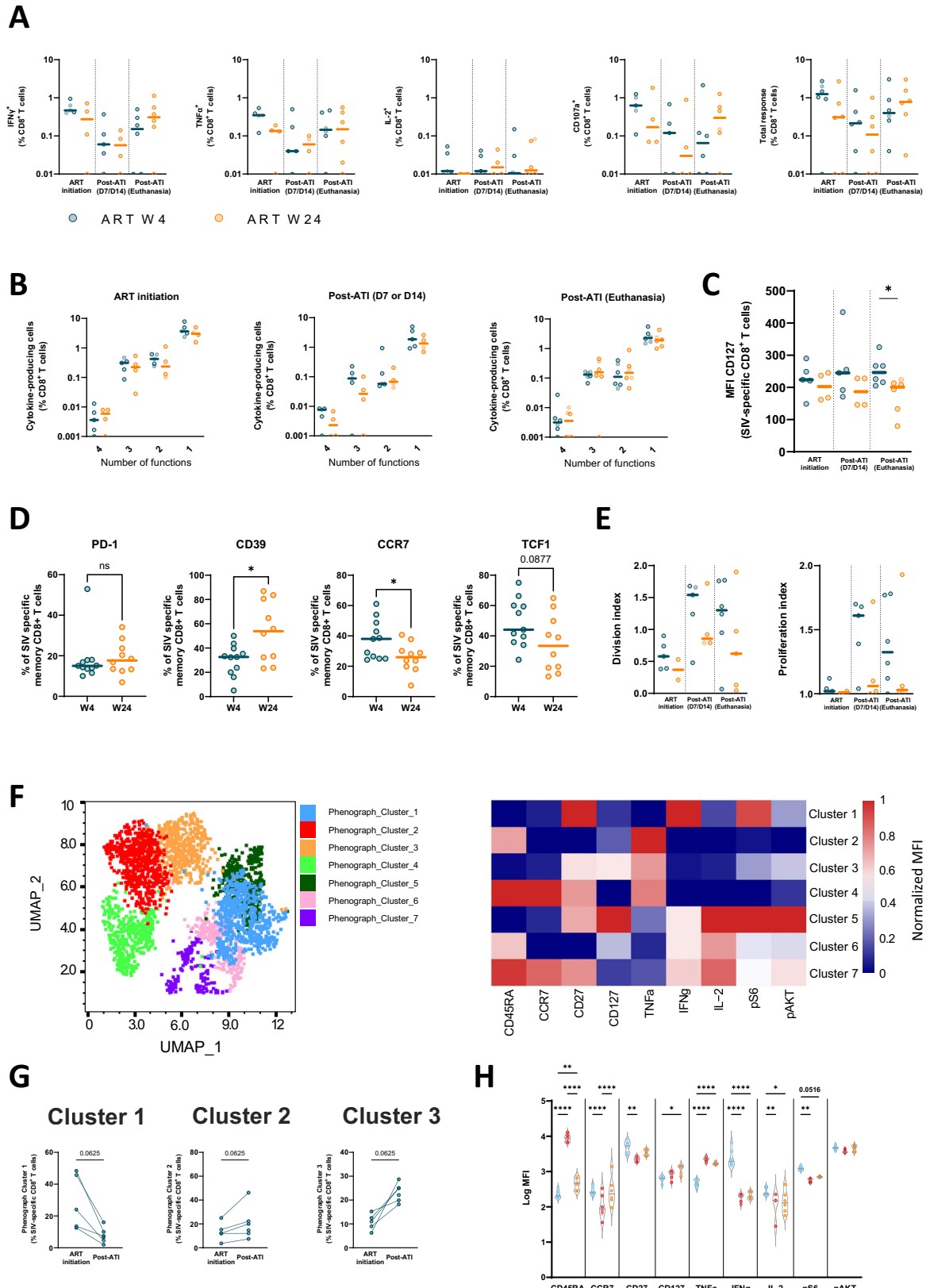

and cell-associated SIV-DNA and CD4⁺ T-cell levels in blood and PLNs. Late-treated animals were exposed longer to uncontrolled viremia and tended to have lower CD4⁺ T cells at the time of ART initiation. However, after two years of ART, no difference that could explain their different outcomes was observed in residual viremia, SIV-DNA levels, cell-associated SIV RNA levels, CD4⁺ T-cell counts or the CD4/CD8 ratio

between early- and late-treated animals. Notably, treatment initiation at 24 weeks post-infection is still relatively early, particularly when considering that ART is initiated years after infection in most PLWH, which may contribute to explaining the normalization of these parameters during ART in both groups of animals. No difference was observed at the time of ART interruption when SIV PTCs were

**Fig. 7 | SIV-specific CD8⁺ T cells with memory-like characteristics are mobilized after interruption of early treatment. A** INFγ, TNFα, IL-2, and CD107a production by SIV-specific CD8⁺ T cells and the total SIV-specific response in the blood of W4-treated and W24-treated CyMs at the time of ART initiation and post-ATI. The results are shown as proportions among CD8⁺ T cells. Negative responses were given an arbitrary value of 0.01. **B** Frequencies of SIV-specific CD8⁺ T cells with 1-to-4 functions based on the expression of IFNγ, TNFα, IL-2, and/or CD107a in W4- and W24-treated CyMs at the time of ART initiation and post-ATI. Negative polyfunctional responses are indicated as 0.001. **C** CD127 expression levels in SIV-specific CD8⁺ T cells at the time of ART initiation and post-ATI. The results are shown as the median fluorescence intensity in SIV-specific CD8⁺ T cells. **D** Proportion of SIV-specific memory CD8⁺ T cells expressing PD-1, CD39, CCR7 or TCF1 in the spleen at the end of the study. **E** Division index (left) and proliferation index (right) of CD8⁺ T cells at the time of ART initiation and post-ATI. **F** Left panel: UMAP plot of n = 2979 SIV-specific CD8⁺ T cells from the blood of W4- and W24-treated CyMs at the time of ART initiation and post-ATI. Phenotypically distinct clusters defined by Phenograph are represented with different colors. Right panel: Heatmap showing the relative expression of differentiation markers, cytokine secretion and markers of mTORC1 (pS6) and mTORC2 (pAKT) pathway activation for each phenotypically distinct cluster shown in (**F**). **G** Dynamics of Clusters 1, 2 and 3 at the time of ART initiation and post-ATI in W4-treated CyMs. **H**) Violin plots comparing the median fluorescence intensity (MFI) of the differentiation markers, cytokine secretion and markers of mTORC1 and mTORC2 pathway activation among Clusters 1, 2 and 3 in W4-treated CyMs; one-way ANOVA, *$p < 0.05$, **$p < 0.01$; ****$p < 0.0001$. **A–E, G, H** Individual data are shown. **A–C, E, G, H** $n = 6$ animals per group. **D** $n = 12$ animals per group. **A–E** Medians are shown; *$p < 0.05$, Two-sided Mann–Whitney U test. **G** Two-sided Wilcoxon matched-pairs rank test. Source data are provided as a Source Data file.

compared to non-PTCs, independent of the delay in ART initiation. Our study nevertheless showed a favorable evolution of post-treatment control, illustrated by low levels of cell-associated SIV-DNA durably maintained after ATI and the preservation of the CD4⁺ T-cell compartment, including a higher frequency of CM CD4⁺ T cells, in blood and lymphoid tissues. These features recapitulate some characteristics of PLWH with durable remission of infection. In contrast, most non-PTCs showed symptomatic progression within months of ART interruption.

Differences in the outcome of early- and late-treated macaques could be related to changes in the immune responses that were mobilized to counteract viral rebound. Regarding the humoral immune response, we found that anti-Env antibody levels during primary infection were linked to the degree and duration of viral exposure as described in human PTCs[51] but that all rebounding animals exhibited potently increased anti-gp140 IgG/IgA titers within days of viral rebound. The overall magnitude of the IgG response was more pronounced for the animals treated at week 24 p.i., and this could be the result of a combination of a higher frequency of persisting memory B cells primed before ART and superior antigenic stimulation. In contrast, no differences were observed for anti-gp140 IgA titers, globally lower than for IgGs, between early- and late-treated animals after ATI, which may be related to a lower frequency of IgA-memory B cells triggered upon primary infection or with a shorter half-life compared to IgG-memory cells. The quantitative changes in anti-SIV antibody levels observed here are likely linked to the different antigenic stimulation between the groups and may not be directly associated with a protective role. Nonetheless, the neutralizing and Fc-dependent antiviral activities of these antibodies will need to be evaluated to better understand their functional role. Indeed, control of viral rebound has been recently related to the presence of antibodies with neutralizing and Fc-dependent effector activities in some early-treated post-treatment controllers[51–53].

Finally, our study suggests that CD8⁺ T cells with enhanced antiviral potential may participate in controlling viremia after ART interruption. Although early-induced CD8⁺ T-cell responses contribute to partially decreased viremia and their pressure is reflected by the emergence of escaping viral variants, numerous reports have pointed to the skewed nature and limited cytotoxic capacity of these cells[45,54–56]. We reported the lack of viral suppressive capacity ex vivo of CD8⁺ T cells isolated during primary infection from PLWH or SIV-infected macaques[36,43]. We previously found that this activity gradually increased over time only in macaques in which viremia was efficiently controlled, namely, those becoming SIV controllers[36]. Moreover, a strong capacity of CD8⁺ T cells ex vivo to suppress viral infection of autologous CD4⁺ T cells is a characteristic that we have consistently found in HIV-1 controllers and in HIV-2 controllers[40,41]. The more frequent occurrence of CD8+ T cells with a strong capacity to suppress infection in the context of SIV and HIV-2 infection suggests that viral factors, for instance, allowing better sensing and T-cell priming of

SIV/HIV-2, may favor their development[57–59]. Here, we show that CD8⁺ T cells from PTC CyMs also showed such a viral suppressive capacity upon ART interruption, enhancing their capacity to counteract viral rebound. The mechanisms underlying this acquisition of the capacity of CD8⁺ T cells to suppress infection remain to be determined but appear to be favored by early ART initiation and could be related to an enhancement of the cytotoxic potential of the cells, an increase in TCR avidity and/or better capacity of CD8⁺ T cells to survive and proliferate in the presence of infected cells[60–62].

Of note, we found that CD8+ T cells expanding in response to primary infection mainly had an effector memory or effector phenotype, while CD8+ T cells expanding after treatment interruption in the early-treatment group had a central memory phenotype. These results suggest that early and prolonged treatment favored the establishment of central memory CD8+ T cells, which mediated a more efficient, secondary-like response against viral rebound. In contrast, cells expanding in the W24-treated animals after ART interruption still carried mostly an effector profile. We and others have recently associated efficient CD8⁺ T-cell responses in HIV and SIV controllers with the development of memory-like CD8⁺ T cells with stem cell properties[36,63,64]. While virus-specific CD8⁺ T cells developed during primary infection do not upregulate molecules associated with long-term memory[36,44], our results in the present pVISCONTI study indicate that early ART introduction may favor the generation/preservation of such cells. These results are consistent with a recent report on early-treated PLWH[25]. Here, CD8⁺ T cells with long-term memory traits and enhanced proliferative and SIV suppressive capacity mediated a robust response to viral rebound while restraining T-cell activation. These results suggest that immunotherapies aimed at the induction of HIV-specific CD8+ T cells with stem cell-like memory properties might be a promising opportunity to achieve HIV remission.

Our study has some limitations. This study was not designed to evaluate the effectiveness of the 2-year duration of ART that we used, or to precisely identify a window of opportunity for ART initiation that would favor post-treatment control of infection. Previous studies in SIV-infected macaques suggest that this period might start with the first signs of immune activation[35] and last at least several weeks[38]. Although the frequency of cells carrying SIV-DNA did not differ between W4- and W24-treated animals at the time of ART interruption in this study, we cannot exclude that early ART initiation impacted the quality, distribution or composition of the viral reservoir, particularly in tissues that were not analyzed here. Many human PTCs appear to consistently maintain undetectable viral loads after ART interruption[13], while others experience transient episodes of viremia[6,11,13,51,65]. In our model, all SIV PTCs except one experienced viral rebound after ATI and may better recapitulate an active control of infection after ART interruption. Some determinants of viral control may differ between macaques and human PTCs. For instance, some immunogenetic traits that have been proposed to favor post-treatment HIV control in PLWH[66] may be absent in CyMs[67]. We did not study the impact of

innate immune responses (IFN-I, dendritic cells, monocytes, NK cells, etc.) in PTCs. It is not excluded that they play a complementary role in posttreatment control, either by impacting CD8+ T-cell function and/or through their antiviral activities. Additional analyses are ongoing to study the impact of early- and late-ART initiation on the innate immune response. Nevertheless, our study showed that post-treatment control, as reported for PLWH[13,16], was not related to the presence of protective MHC traits associated with natural control of infection (e.g., HLA-B*27 or B*57 alleles in PLWH or MHC-M6 haplotype in CyMs).

In summary, we formally demonstrate in our model a direct role of ART initiation during primary infection in the promotion of post-treatment SIV control. While the CyM model might not recapitulate all traits present in humans, further studies using this model should provide new insights into the virological and immunological events leading to post-treatment control of infection. Of note, our analyses identified key features in SIV-specific CD8+ T cells shaped under early- vs. late-ART that may underlie a different capacity to respond to viral rebound. These results thus pave the way for improving CD8 + T-cell-based immunotherapies toward remission of HIV.

## Methods

### Ethical statement

CyMs were imported from Mauritius and housed in facilities at the Infectious Disease Models and Innovative Therapies (IDMIT) center (CEA site in Fontenay-aux-Roses, France). All nonhuman primate studies at IDMIT were conducted in accordance with French national regulations under the supervision of national veterinary inspectors (CEA Permit Number D92-032-02). IDMIT complies with the standards for Human Care and Use of Laboratory Animals of the Office for Laboratory Animal Welfare under Assurance Numbers #A5826-01 and F20-00448. All experimental procedures were conducted according to European Directive 2010/63 (Recommendation Number 9). The pVIS-CONTI study was approved and accredited under the statement A15 035 from the "Comité d'Ethique en Expérimentation Animale du CEA" and was registered and authorized under Number 2453-2015102713323361v3 by the French Ministry of Education and Research. Experimental procedures (animal handling, viral inoculations, and samplings) were conducted after sedation with ketamine chlorhydrate (Imalgene 1000®, 10 mg/kg, intravenously (i.v.), Merial). Tissues were collected during follow-up and at necropsy. Animals were euthanized after ketamine chlorhydrate sedation followed by a bolus of sodium pentobarbital (Doléthal, 180 mg/kg, i.v., Laboratoire Vetoquinol).

### Animals and experimental conditions

This stage of the pVISCONTI study included a total of 41 male CyMs (median age = 4.8 years at inclusion, IQR = 3.9–7.2). CyMs were inoculated intravenously (i.v.) with uncloned SIVmac251 at an animal infectious dose (AID)50 of 1000. SIVmac251 stock was kindly provided by Anne-Marie Aubertin (Université Louis Pasteur, Strasbourg, France)[68]. Briefly, this stock was obtained from the cell-free supernatant of infected rhesus PBMCs, which had been infected in vitro with a culture supernatant obtained from a coculture of rhesus PBMCs and a spleen homogenate from a rhesus macaque infected with SIVmac251 (provided by R. C. Desrosiers, New England Regional Primate Center, Southborough, Mass).

These experimental conditions ensure homogenous high plasma viremia during primary infection and do not favor spontaneous SIV control[69]. ART containing emtricitabine (FTC, 40 mg/kg, Gilead), dolutegravir (DTG, 2.5 mg/kg, ViiV Healthcare), and the tenofovir prodrug tenofovir-disoproxil-fumarate (TDF, 5.1 mg/kg, Gilead) coformulated as a once daily subcutaneous injection was initiated either at week 4 (12 animals) or at week 24 (12 animals) post-infection (p.i.). CyMs received ART for 24 months, which was further interrupted. Animals were then monitored for viral rebound (first viral load

>400 SIV RNA copies after ART interruption) and/or post-treatment SIV control. Animals were classified as PTCs if achieved at least one pVL of <400 SIV RNA copies/ml after viral rebound (or if they never experienced viral rebound). The 400 RNA copies/ml threshold mirrored that of HIV PTCs in the VISCONTI study[13]. The prescheduled end of the study was 48 weeks post-treatment interruption. However, most animals without post-treatment control progressed to clinical scores requiring euthanasia before this time. Nevertheless, all animals were monitored for at least 6 months after treatment interruption (Fig. S1, Table S1).

Of note, this stage of the pVISCONTI study was performed in two independent experimental phases, each including 6 early- and 6 late-treated macaques that were infected, treated, and monitored in parallel (for detailed study protocol, see Fig. S1). In addition, a total of 17 untreated CyMs in three experimental phases ($n$ = 5, 6 and 6) were analyzed throughout the study and were used as a reference for natural evolution under our experimental conditions (Table S1). The different groups of animals studied were matched for age and MHC haplotypes. CyMs harboring the M6 haplotype, previously described to favor natural SIV control[36,70,71], were not included in the study. The characteristics of the CyMs in this study are detailed in Table S1. One animal (CB296A) presented an extremely severe acute infection and reached the endpoint requiring euthanasia before ART initiation and was excluded from further analyses. Some analyses of the CD8+ T-cell response were limited to the animals from the pVISCONTI-1 experiment ($n$ = 6 animals in each group), in which samples were specifically destined for deeper immunological characterization.

### Blood collection and processing

Peripheral blood was collected by venous puncture into Vacutainer Plus Plastic K3EDTA Tubes or Vacutainer CPT Mononuclear Cell Preparation Tubes with Sodium Heparin (BD Biosciences). Complete blood counts were monitored at all time points from the Vacutainer Plus Plastic K3EDTA Tubes. Plasma was isolated either from Vacutainer Plus Plastic K3EDTA Tubes by centrifugation for 10 min at 480 × $g$ or from Vacutainer CPT Mononuclear Cell Preparation Tubes with Sodium Heparin after centrifugation for 40 min at 1900 × $g$, and the samples were stored at −80 °C. Peripheral blood mononuclear cells (PBMCs) were isolated from Vacutainer CPT Mononuclear Cell Preparation Tubes with Sodium Heparin according to the manufacturer's instructions (BD Biosciences), and red blood cells were lysed in ammonium-chloride-potassium (ACK) buffer (0.15 M NH4Cl, 10 mM KHCO3, 0.1 mM EDTA, pH 7.4).

### Tissue collection and processing

Axillary or inguinal lymph node (peripheral PLN), bone marrow (BM), and bronchoalveolar lavage (BAL) samples were collected longitudinally (Fig. S1). In addition, the spleen, mesenteric lymph nodes (MLNs), colon and liver were collected at necropsy. Tissue samples were collected in RPMI medium at 2–8 °C. Lymph node cells were isolated into RPMI medium via mechanical disruption using a gentleMACS Dissociator (Miltenyi Biotec). The cell suspension was filtered (70 μm), and red blood cells were lysed in ACK. Bone marrow cells were purified using Lymphocyte Separation Medium (Eurobio Scientific) diluted to 90% in DPBS, centrifuged for 20 min at 350 × $g$, and separated from red cells in ACK. Spleen cells were processed via mechanical disruption in RPMI medium using a gentleMACS Dissociator (Miltenyi Biotec), purified as described for BM cells, and separated from red cells in ACK. Colonic lymphocytes were obtained from mucosa taken from approximately 10 cm of tissue. Colonic tissue was washed extensively in PBS and R10 medium (RPMI medium supplemented with 10% fetal calf serum and penicillin/streptomycin) and then digested for 45 min with collagenase II prior to mechanical disruption. Lymphocytes were isolated over a Percoll 67/44 gradient (Sigma-Aldrich). Liver tissue was mechanically disrupted using a gentleMACS Dissociator (Miltenyi

Biotec), and the cell suspension was sequentially filtered (300 μm > 100 μm > 70 μm). Then, lymphocytes were obtained over an OptiPrep gradient (Sigma-Aldrich). T-cell activation, proliferation and exhaustion phenotyping and measurements of SIV-suppressive activity ex vivo were performed using freshly isolated cells. T-cell CFSE proliferation assays and cytokine intracellular staining were performed using viable cells frozen at −196 °C in DMSO/FCS.

## Quantification of plasma viral load
Plasma viremia was monitored longitudinally in all animals using quantitative RT-PCR with a limit of detection of 12.3 copies/mL. Viral RNA was prepared from 100 μL of cell-free plasma. Quantitative RT-PCR was performed using a SuperScript III Platinum One-Step qRT-PCR Kit (Thermo Fisher Scientific) with a CFX96 Touch Real-Time PCR Detection System (Bio-Rad). Each tube contained 12.5 μL of 2× reaction mixture, 0.5 μL of RNaseOUT (40 U/μL), 0.5 μL of Superscript III Reverse Transcriptase/Platinum Taq DNA Polymerase, 1 μL of each primer (125 μM), 0.5 μL of the fluorogenic probe (135 μM), and 10 μL of eluted RNA. Primer/probe sequences were designed to amplify a region of the SIVmac$_{251}$ *gag* gene. The forward (F) primer sequence was 5′-GCAGAGGAGGAAATTACCCAGTAC-3′ (24 bp), and the reverse (R) primer sequence was 5′-CAATTTTACCCAGGCATTTAATGTT-3′ (25 bp). The probe sequence was 5′-FAM-TGTCCACCTGCCATTAAGCCCGA-BHQ1-3′ (23 bp). This probe had a fluorescent reporter dye, FAM (6-carboxyfluorescein), attached to its 5′ end and a quencher, BHQ1 (black hole quencher 1), attached to its 3′ end (TaqMan, Applied Biosystems). Samples were heated for 30 min at 56 °C and 5 min at 95 °C, followed by 50 thermocycles, each comprising 15 sec at 95 °C and 1 min at 60 °C.

## Quantification of SIV-DNA and cell-associated SIV-RNA
Total DNA and caRNA were extracted from PBMCs with an AllPrep DNA/RNA Mini Kit (Qiagen). For tissue, to control for differences in viral distribution within a single organ, two or three tissue samples were first mechanically disrupted separately with a MagNA Lyser (GmbH, Roche Diagnostics, Brussels, Belgium). Nucleic acids were extracted separately, and the lysate was divided into two parts for DNA (DNA Mini Kit, Qiagen) and RNA extraction (RNeasy Plus Mini Kit, Qiagen). caRNA was treated during extraction by DNase I (Qiagen). Then, viral measurements were performed on each extract.

To determine the infection burden, total SIV-DNA was quantified by ultrasensitive real-time PCR targeting the gag region[72]. The cycling conditions were 2 min at 50 °C and 10 min at 95 °C followed by cycles at 95 °C for 15 s and 60 °C for 1 min. The limit of detection was 1 copy/PCR. To normalize SIV-DNA per million cells, the CCR5 gene was quantified by real-time PCR with the same cycling conditions described above (forward primers: an equimolar mix of the primers 5′-CAACATG CTGGTCGATCCTCAT-3′ and 5′-CAACATACTGGTCGTCCTCA TCC-3′, reverse primer: 5′-CAGCATAGTGAGCCCAGAAG-3′, probe: 5′-HEX-CTG ACA TCTACCTGCTCAACCTG-BHQ1-3′). For both PCRs, the reaction volumes were 50 μL, containing 25 μL of 2× qPCR Mastermix Plus (Eurogentec, Seraing, Belgium), 0.4 μM each primer and 0.2 μM probe. SIV-DNA levels were reported as copies per $10^6$ cells or CD4$^+$ T cells or as threshold values when the SIV-DNA level was under the detection threshold or SIV-DNA was not detected.

SIV caRNA was quantified by one-step real-time PCR targeting the gag[72]. Two replicates of all amplifications were performed. The cycling conditions were 10 min at 50 °C and 5 min at 95 °C followed by cycles at 95 °C for 15 s and 60 °C for 1 min. The limit of detection was 5 copies/ PCR. Normalization of SIV caRNA was performed with ribosomal RNA (one-step real-time PCR using the Ribosomal RNA Control Reagents Kit (Applied Biosystems)). The reaction volume was 20 μL, containing 5 μL of 4× Master Mix Fast Virus 1-Step Taqman (Applied Biosystems), 0.4 μM each primer and 0.2 μM probe.

A standard using SIV1C cells containing a single integrated copy of SIVmac$_{251}$ proviral DNA per cell was used (https://www.resourcenhpir. com/data-sheet-siv-1c-cell-line)[73]. The DNA and RNA thresholds varied according to the number of cells and the quantity of total RNA available and were calculated for each assay.

## Titration of SIVmac251 gp140 IgG and IgA
SIVmac$_{251}$ g140-foldon-type glycoproteins were produced by transient transfection of FreeStyle™ 293-F cells and purified by affinity chromatography using Ni Sepharose® Excel beads (GE Healthcare)[74]. High-binding 96-well ELISA plates (Costar, Corning) were coated overnight with 250 ng/well of purified recombinant SIV gp140 protein. After washing with 0.05% Tween 20-PBS (washing buffer), plates were blocked for 2 h with 2% BSA, 1 mM EDTA, and 0.05% Tween 20-PBS (blocking buffer), washed, and incubated with serially diluted NHP sera in duplicate at 1:250 or 1:50 followed by 7 consecutive 1:4 or 1:3 dilutions in PBS for IgG or IgA detection, respectively. After washing, the plates were revealed by incubation for 1 h with goat HRP-conjugated anti-human IgG or IgA antibodies (Jackson ImmunoReseach, 0.8 μg/ml final) and by adding 100 μl of HRP chromogenic substrate (ABTS solution, Euromedex). Optical densities were measured at 405 nm (OD405 nm), and background values based on incubation of PBS alone in coated wells were subtracted. Experiments were performed using a HydroSpeed™ microplate washer and Sunrise™ microplate absorbance reader (Tecan Männedorf, Switzerland).

## Measurement of SIV-suppressive activity
Autologous CD4$^+$ and CD8$^+$ T cells were purified from freshly isolated PBMCs or tissue cell suspensions by positive and negative selection, respectively, using nonhuman primate CD4$^+$ MicroBeads and the CD8$^+$ T-Cell Isolation Kit, with the MultiMACS™ Cell24 Separator (Miltenyi Biotec). Purified CD4$^+$ T cells were stimulated for 3 days with concanavalin A (5 μg/mL, Sigma-Aldrich) in the presence of IL-2 (100 IU/ mL, Miltenyi Biotec). Purified CD8$^+$ T cells were cultured in the absence of mitogens and cytokines (ex vivo CD8$^+$ T cells). Stimulated CD4$^+$ T cells ($10^5$) were superinfected in U-bottom 96-well plates with SIVmac$_{251}$ (MOI = $10^{-3}$) in the presence (1:1 effector-to-target-cell ratio) or absence of ex vivo CD8$^+$ T cells ($10^5$) from the same tissue via spinoculation[75] for 1 h (1200 × g at room temperature) and incubated for 1 h at 37 °C. Cells were then washed and cultured in R10 medium containing IL-2 (100 IU/mL, Miltenyi Biotec). Culture supernatants were assayed on day 7 using an SIV p27 Antigen ELISA Kit (XpressBio). Antiviral activity was calculated as log10 (mean p27 ng/mL in SIV-infected CD4$^+$ T-cell cultures without ex vivo CD8$^+$ T cells)/(mean p27 ng/mL in SIV-infected CD4$^+$ T-cell cultures with ex vivo CD8$^+$ T cells)[39].

## Analyses of T-cell activation and proliferation markers
Total T-cell activation and proliferation were assessed using fresh PBMCs and tissue cell suspensions. Cells were stained with the LIVE/DEAD Fixable Aqua Dead Cell Stain Kit (Thermo Fisher Scientific) and then surface stained for CD3, CD4, CD8, CD38, and HLA-DR. For differentiation phenotype analyses, cells were additionally stained with CD45RA, CCR7 and CD27. Then, the cells were fixed/permeabilized using a Cytofix/ CytoPerm Kit (BD Biosciences) and stained for intracellular Ki-67. The following antibodies were used: anti-CD3−AF700 (clone SP34-2, BD Biosciences), anti-CD4−PerCP-Cy5.5 (clone L200, BD Biosciences), anti-CD8−APC-Cy7 (clone RPA-T8, BD Biosciences), anti-CD38−FITC (clone AT-1, StemCell Technologies), anti-HLA-DR−V450 (clone G46-6, BD Biosciences), anti-CD45RA−PE Cy7 (clone 5H9, BD Biosciences), anti-CCR7−PE-Dazzle594 (clone G043H7, Biolegend), anti-CD27−PE (clone M-T271, BD Biosciences), and anti-Ki-67− AF647 (clone B56, BD Biosciences). Data were acquired using an LSRII flow cytometer (BD Biosciences) and analyzed with FlowJo software version 10 (Tree Star Inc.). Cells were classified as follows: naive−CD45RA$^+$CD27$^+$CCR7$^+$; central

memory (CM)−CD45RA⁻CD27⁺CCR7⁺; transitional memory (TM)−CD45RA⁻CD27⁺CCR7⁻; effector memory (EM)−CD45RA⁻CD27⁻CCR7⁻; and effector−CD45RA⁺CD27⁻CCR7⁻ (Fig. S2).

## Cytokine production, memory and proliferative capacity of SIV-specific T cells

Frozen PBMCs were thawed, resuspended at $1 \times 10^6$/mL in R20 medium, and incubated overnight at 37 °C. Cells were then stimulated with a pool of 24 optimal SIV peptides (8–10 amino acids, 2 µg/mL each, Table S2[36]) in the presence of anti-CD28 (1 µg/mL, clone L293, BD Biosciences) and anti-CD49d (1 µg/mL, clone 9F10, BD Biosciences) and stained with anti-CD107a (clone H4A3, BD Biosciences) for 30 min prior to the addition of GolgiStop (1 µL/mL, BD Biosciences) and brefeldin A (BFA, 5 µg/mL, Sigma-Aldrich). Costimulatory antibodies alone were used as a negative control, and concanavalin A (5 µg/mL, Sigma-Aldrich) was used as a positive control. Cells were incubated for a total of 6 hr. To evaluate the differentiation profile, cells were stained with the LIVE/DEAD Fixable Aqua Dead Cell Stain Kit (Thermo Fisher Scientific) and then surface stained for CD3, CD4, CD8, CD45RA, CCR7, CD27 and CD127. Cells were then fixed/permeabilized using the Cyto-fix/CytoPerm Kit (BD Biosciences) and stained intracellularly for IFNγ, TNFα, and IL-2 (Fig. S3). The results were corrected for background by subtracting the negative (no peptide) control from the peptide-stimulated response. To evaluate the memory profile, cells were stained with the LIVE/DEAD Fixable Aqua Dead Cell Stain Kit (Thermo Fisher Scientific) and then surface stained for CD3, CD8, CD45RA, CCR7, CD27, CD39, and PD-1 in brilliant stain buffer (BD Bioscience). Cells were then fixed/permeabilized using Phosflow buffer (BD Biosciences) and stained intracellularly for TCF-1, IFNγ and TNFα (Fig. S4).

CD8⁺ T-cell proliferation upon stimulation was evaluated by CFSE staining (1 µM, Thermo Fisher Scientific) (Fig. S5). PBMCs were CFSE stained, stimulated with a pool of 24 optimal SIV peptides (2 µg/mL each), and then cultured for 6 days. Twelve hours before completing the total incubation time, an additional dose of the peptides was added as well as GolgiStop (1 µL/mL, BD Biosciences), brefeldin A (BFA, 5 µg/mL, Sigma-Aldrich), and an anti-CD107a antibody. Unstimulated cells were used as a negative control. Cells were stained with the LIVE/DEAD Fixable Aqua Dead Cell Stain Kit (Thermo Fisher Scientific) and then surface stained for CD3, CD4, CD8, CD45RA, CCR7, CD27 and CD127. Cells were then fixed/permeabilized with PhosFlow fix/perm buffers (BD Biosciences) and stained intracellularly for IFNγ, TNFα, IL-2, pS6 Ser235/236 and anti-pAKT Ser473.

The following antibodies were used: anti-CD3−AF700 (clone SP34-2, BD Biosciences), anti-CD4−PerCP-Cy5.5 (clone L200, BD Biosciences), anti-CD8−APC-Cy7 (clone RPA-T8, BD Biosciences), anti-CD45RA−PE Cy7 (clone 5H9, BD Biosciences), anti-CCR7−PE-Dazzle594 (clone G043H7, Biolegend), anti-CD27−PE (clone M-T271, BD Biosciences), anti-CD127−BUV496 (clone HIL-7R-M21, BD Biosciences), anti-CD107a−BV786 (clone H4A3, BD Biosciences), anti-TNFα−BUV395 (clone MAb11, BD Biosciences), anti-IFNγ−BV605 (clone B27, BD Biosciences), anti-IL2−BUV737 (clone MQ1-17H12, BD Biosciences), anti-phospho S6 S235/236−Pacific blue (clone D57.2.2E, Cell Signaling), anti-phospho Akt Ser473−AF647 (clone D9E, Cell Signaling), anti-CD279 (PD1)−BV 421 (clone EH12.2H7, Biolegend), anti-CD39−BV785 (clone A1), and anti-TCF-7/TCF-1−FITC (clone S33-96, BD Bioscience). Data were acquired using a Fortessa flow cytometer (BD Biosciences) and analyzed with FlowJo software version 10 (Tree Star Inc.). The results were corrected for background by subtracting the negative (no peptide) control from the peptide-stimulated response. Negative responses were given an arbitrary value of 0.01. Division and proliferation index calculations were performed using the proliferation modeling function available in FlowJo. The division index is the average number of cell divisions that a cell in the original population has undergone. The proliferation index is the total number of divisions

divided by the number of cells that went into division. Clustering analysis of the SIV-specific CD8⁺ T-cell dataset was performed on concatenated files from early- and late-treated CyMs at the time of ART initiation and post-ATI. Dimensionality reduction was performed using the uniform manifold approximation and projection (UMAP) algorithm (FlowJo plugin v3.1). UMAP was performed using the following parameters: metric−Euclidian, nearest neighbors−15, minimum distance−0.5, and number of components−3. Clustering analysis was performed using the unsupervised PhenoGraph algorithm (FlowJo plugin v2.4). PhenoGraph was used for unsupervised clustering using default parameters (K = 30). All markers were considered for UMAP and PhenoGraph analyses, except live/dead and CD4. The median fluorescence intensity (MFI) for each cluster was normalized through the min−max scaling method and can be visualized in the heatmap (R version 4.2.2) (Fig. S6).

## Data visualization and statistical analyses

Data storage was ensured with the BaTLab laboratory management system of IDMIT, and database interrogation and data visualization were performed using Tableau version 2021.3 (Tableau Software). Graphs and statistical analyses were performed using Prism version 9.2.0 (GraphPad Software) or R version 4.2.2 (http://www.R-project.org). The results are presented as individual data with the median. Groups were compared using the Mann–Whitney U test. The Friedman test and Wilcoxon matched-pairs rank test were used to compare paired samples. Tests were two-sided. Correlations were assessed using Spearman rank analyses. No adjustments were made for multiple comparisons, given the exploratory nature of the analyses. All p values less than 0.05 were defined as significant.

## Reporting summary

Further information on research design is available in the Nature Portfolio Reporting Summary linked to this article.

# Data availability

The data that support the findings of this study are presented in the main figures and supplemental material of this article. Further information and requests for resources and reagents should be directed to the lead contact. Request for biological resources will be fulfilled based on availability and upon the establishment of an MTA. Source data are provided with this paper.

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

## Acknowledgements

The pVISCONTI study was funded by MSDAvenir, through a research grant to the ANRS-RHIVIERA consortium, and the ANRS | Emerging Infectious Diseases French agency (ANRS-MIE). Additional support was provided by the Programme Investissements d'Avenir (PIA), managed by the ANR under reference ANR-11-INBS-0008, funding IDMIT infrastructure. A.C. was supported by an ANRS Postdoctoral Fellowship. F.P.C. was supported by ANRS and a Pasteur-Roux-Cantarini Postdoctoral fellowship from Institut Pasteur. A.S-C and M.G. were supported by UM1AI164562, co-funded by National Heart Lung and Blood Institute, National Institute of Diabetes and Digestive and Kidney Diseases, National Institute of Neurological Disorders and Stroke, National Institute on Drug Abuse, and the National Institute of Allergy and Infectious Diseases. We thank Animalliance and animal care workers, in particular Sebastien Langlois, Benoit Delache, Claire-Maelle Fovet, Maxime Pottier, Jean-Marie Robert, Quentin Sconosciuti, Nina Dhooge, Emma Burban and Julie Morin, Laetitia Bossevot, Brice Targat, Wesley Gros, Marco Leonec, Anne-Sophie Gallouet for expert technical assistance and Isabelle Mangeot-Méderlé for helpful project management at IDMIT. We thank Dr. Geronimo Dubra for bioinformatic analysis support. FTC, DTG, and TDF were obtained from Gilead and ViiV Healthcare through the "*IAS Toward an HIV Cure*" common Material Transfer Agreement. The SIV1C cell line was kindly provided by François Villinger.

## Author contributions

C.P. designed and carried out the experiments, analyzed the data, and interpreted the results. A.C., V.M., J.L., F.P.C., A.M., Cy.P., M.G., A.D., N.D., H.G., C.G. and F.R. carried out experiments, analyzed data, and interpreted results. D.D. and N.D.B. designed the experiments, analyzed the data, and interpreted the results. J.G., H.M. and V.A.F. analyzed the data and interpreted the results. A.B.T., O.L., M.M.T., and C.R. contributed to the study design and interpreted the results. C.R., R.L.G. and A.S.C. obtained funding. R.L.G. and A.S.C. designed the experiments, analyzed the data, interpreted the results, and supervised the study. C.P. and A.S.C. wrote the manuscript. All authors critically reviewed the manuscript and contributed to the final version.

## Competing interests

A.S.C. has received speaker fees from MSD, ViiV Healthcare, Gilead, Janssen. V.A.-F. has received grants (to her institution) from ViiV Healthcare and honoraria and travel grants from ViiV Healthcare and Gilead Sciences for participation in educational programs and conferences. The other authors declare no competing interests.

## Additional information

[1]Institut Pasteur, Université Paris Cité, Viral Reservoirs and Immune Control Unit, Paris, France. [2]Institut Pasteur, Université Paris Cité, HIV Inflammation and Persistence Unit, Paris, France. [3]Université Paris-Saclay, CEA, INSERM, UMR1184, Immunology of Viral, Auto-immune, Hematological and Bacterial diseases (IMVA-HB/IDMIT Department), Fontenay-aux-Roses/Le Kremlin-Bicêtre, France. [4]Université Paris Cité; INSERM, U1016; CNRS, UMR8104 Paris, France. [5]Institut Pasteur, Université Paris Cité, INSERM U1222, Humoral Immunology Unit, Paris, France. [6]Université Paris-Saclay, AP-HP, Hôpital Bicêtre, Service de Pharmacie, Le Kremlin Bicêtre, France. [7]Université Paris-Saclay, AP-HP. Hôpital Bicêtre, Clinical Immunology Department, 94270 Le Kremlin Bicêtre, France. [8]Université Paris Cité, IAME, INSERM, F-75018 Paris, France. [9]Université Paris Cité/APHP Hôpital Necker - Enfants Malades, Paris, France. [10]APHP Hôpital Cochin, Service de Virologie, Paris, France. [11]These authors contributed equally: Roger Le Grand, Asier Sáez-Cirión. ✉e-mail: cpereira@pasteur.fr; asier.saez-cirion@pasteur.fr

