## [Peer Review File · Nature Communications]

REVIEWER COMMENTS

Reviewer #1 (Remarks to the Author):

In this study, Passaes et al. utilize SIVmac251-infected cynomolgus macaques that received 2 years of ART started at week 4 (early) or week 24 (late) post infection and that underwent analytic treatment interruption (ATI). This design allowed them to determine if and how time of ART initiation impacts on promoting post-treatment control (PTC) in absence of ART. Furthermore, they performed extensive virologic and immunologic analyses to determine potential correlates and mechanisms of PTC. The same group characterized for the first time, now almost 10 years ago, post-treatment control in a very limited number of people living with HIV that underwent ATI, thus providing evidence that HIV remission can be achieved in a subset of people after ART discontinuation. However, the mechanisms responsible for viral remission remain unclear. As such, creating a NHP model to study PTC is of high significance for the HIV field, allowing to determine the factors involved in PTC using homogeneous experimental conditions; to perform extensive analyses of the virologic and immunologic features of control, including at pre-infection and pre-ART; to expand the analyses to tissues, including secondary lymphoid tissues that have been indicated as critical for viral replication and persistence.

The study was very well designed – including maintaining animals on ART for two years – and performed. The manuscript is very well written and includes a large amount of work convincingly showing that early ART strongly promote durable PTC in SIV-infected cynomolgus macaques and that this control was associated with the generation/expansion of long-term memory CD8 T cells with enhanced antiviral activities. These findings are very important and significant.

I do have few comments to further improve the quality of the manuscript, as listed below:

Main Comments:

1. Since control is associated with the generation/expansion of long-term memory CD8 T cells, it will be interesting to further characterize those cells, including for the expression of immune checkpoint inhibitors and other markers that have been associated with T cell function, differentiation status, and longevity (for example PD-1, Tox, TCF-1, CD39).
2. The authors show a quite impressive difference among the two groups in the levels of CM CD8 T cells in the colon and speculate this can be due to different levels of gut integrity, inflammation, and microbial translocation among the early and late ART groups. This concept can be tested by measuring soluble markers of inflammation/gut integrity (LPS, LBP, sCD14, etc.) and/or levels of IL-17 producing CD4 T cells, that has been indicated as critical for gut integrity.

3. The study very convincingly shows different level of IL-7R expression in SIV-specific CD8 T cells among the two groups. Are the levels of IL-7 different at ATI and post ATI?

Minor Comments:

1. Were the levels of the CD8 T cell differentiation subsets (SCM, CM, EM) different among the two groups at ATI?
2. The authors showed an interesting correlation between the expansion of CM CD8 T cells and the suppressive activity of CD8 T cells, both in blood and LN. Is there a correlation between the frequency of proliferating (Ki-67+) CM CD8 T cells and the level of CM CD8? What about the frequency of proliferating CM CD8 T cells and the suppressive activity?

Reviewer #2 (Remarks to the Author):

This study design was informed by observational work demonstrating characteristics of PLWH experiencing long term, spontaneous viral control post antiretroviral treatment tended to have early institution of ART followed by at least two years of treatment prior to interruption. Here they recapitulate this model in Cynomolgus macaques with SIVmac251 and compared post-treatment control defined as a persistent VL under 400 copies/mL between treatment approaches distinguished by receiving ART at either 4 weeks post infection or 24 weeks post infection. Untreated NHP were also included. There were multiple experimental groups for intervention and sham recipient likely to accommodate logistic and resource limitations. The groups starting ART at 4 weeks post infection experienced lower rebound viremia off treatment and a high proportion achieved PCT status (81%) versus the sham and late (24 week) treatment initiation groups which experienced higher viral rebound, clinical progression and only 11% and 18% achieved PCT status respectively. This study is notable for its design relative to the time of ART initiation in more clinically realistic settings. The large majority of cure research explores novel interventions for “cure” by treating very early during the era of acute SIV or HIV infection in the belief that reducing the viral reservoir and preserving immune function with very early therapy affords the best opportunity for identifying a successful strategy to reduce the rebound competent reservoir. This study adds to the field notably by demonstrating that initiation of ART later without other intervention may result in a very high proportion of viral controllers post ATI. There are many studies intervening with ART within 3 – 14 days of infection and continuing treatment for 72 – 86 weeks or more. Here are a couple of studies of note:

Borducci et al. *Nature*. 2016 Dec 8; 540(7632): 284–287.

Walker-Sperling et al. *Nat Commun*. 2022; 13: 3463.

These studies observe rapid rebound in controls receiving only ART for extended periods (albeit somewhat shorter than the study under review). The Passaes et al paper discuss plausible explanations for the observed failure of very early treatment initiation to achieve the results reported here. They propose that some interval of persistent antigen exposure is needed to mature and expand the pathogen specific immune response which after 2 years of suppressive treatment is able to mount a protective immune response and though viral rebound occurs, the rebound is subdued and ultimately controlled to a level of 400 copies/mL or less. They argue that later treatment permits establishment of an extensive reservoir and results in immune system damage which reduces the probability of post treatment control. This “Goldilocks scenario” of just the right amount of viremia is appealing but there remains much to explore and refine. I am not sure why the study was performed in Cynomolgus macaques where so many other studies in the field use Rhesus. Further, it would have been useful to include a group with very early intervention at day 7 – 10 to buttress the argument they forward in the discussion, lines 644-651. A critical insight here is that the field should look more formally at timing of interventions despite the ample evidence that treatment during AHI reduces the reservoir and preserves immune integrity. However critically, we need to better understand the viral and immune parameters associated with PCT. Here we have evidence of CD8 suppression of autologous CD4 cell SIV infection associating with the 4 week ART group showing a trend to higher CD8 suppressive activity prior to ATI which increases during the ATI. No other measured markers seem to correlate with viral control though IL 7 expression was higher in the W4 group and among all PCT animals compared to non PCT animals. Taken together the paper demonstrates a dramatic increase in PTC among the W4 group with no evidence of benefit among the W24 ART group compared to no ART groups and W4 ART favored establishment of memory like SIV specific CD8 cells with enhanced proliferative potential and survival capacity. These are important observations but it seems there is much left to analyze which may be far more informative by an in depth analysis of other reservoir measurements in different T cell compartments and additional functional assay work for innate responses and humoral effectors.

Key observations:

- 1) A dose response so to speak with respect to timing of ART initiation: 2 of 17 controls with no ART developing spontaneous control, 2 of 11 starting ART at 24 weeks (W24) and a surprising increase to 9 of 11 receiving ART at 4 weeks (W4)! This is a remarkable model and both encouraging and yet sobering, as these intervals were based on the human experience reported in VISCONTI and suggests that there may be value to early therapy but somewhere before 4 weeks and 24 weeks is the beginning and end of the window of opportunity based on ART intervention alone.
- 2) Reservoir is measured here by total SIV DNA. While it would have been illuminating to use other measures of replication competent provirus (TILDA and IPPD) to better characterize the reservoir. Interestingly, there was greater SIV DNA signal in LN in the 4 week intervention group at 4 weeks than in the 24 week intervention group at 24 weeks and as there were no differences in these groups pre ATI or early after ATI one assumes there was differential decay in the two groups. The 4 week group had lower SIV DNA at study end or euthanasia as expected.

3) Not unexpectedly, CD4 cells were better preserved in the 4 week treatment group post ATI and similar at all other time points.

4) Antibody developed poorly in the 4 week intervention group with loss of antibody formation upon ART initiation. Antibody developed robustly in the 24 week ART group but to a lesser extent than the W4 group, ART suppressed antibody formation. Both groups produced antibody beginning at ATI although week 24 animals had higher antibody at ATI. Antibody levels correlated positively with pVL AUC which is not unexpected.

5) A negative correlation is observed between autologous CD8 suppressive activity and log VL AUC and VL at euthanasia. The 4 week group was not different from the 24 week group for this parameter until pre ATI and this difference grew in magnitude post ATI. In addition, PTC animals had better suppressive activity than non PTC. Association of this CD8 suppressive activity with PTC has been reported by this group but the effector mechanism and the means of inducing this response remains unknown. Suppressive activity is correlated with CM CD8 T cells but not other CD8 T cell subsets.

6) In general, expression of IL-7 was higher in W4 treated macaques post ATI and PTCs in general had higher levels of IL-7 expression but no other differences in polyfunctionality or single cytokine expression were noted. Also overall SIV specific CD8 between PTC and non PTC. However, the proliferation of W4 treated Cyno was higher early post ATI.

Comments:

1) Methods: the laudable and ethical conditions of NHP management in your study are more detailed in lines 96-117 than needed for the reader and can be shortened (not eliminated) with content shifted as appropriate to the supplement.

2) Methods: This sentence starting on line 132 is a bit unclear: Most animals without post treatment control reached clinical scores indicating euthanasia before the pre-scheduled follow up period, but with at least 6 months of follow up after treatment interruption (Figure S1, Tables S1). Please clarify.

3) Methods: It is not state of the art to measure total HIV DNA as TILDA or IPDA measurements better reflect the rebounding, ie replication competent reservoir and given the tissue collections, determining the cell phenotype of reservoir cells might be feasible and very informative as well.

4) Methods: the endpoint definition is noted in the results but indicates PTC required VL <400 copies per mL but not whether this needed to be a durable result over several measurements or just the last study measurement. This should be explicitly stated in the methods as the primary endpoint and key definitions like "rebound" should be included as well.

5) Line 340 references the figure S1C...a critical figure depicting VL over time in the 7 treatment and control experiments (2 groups of W4, 2 groups of W24 and 3 groups with no ART). I am trying to figure out the labels in the figure for W4 and W24 week treatment groups as the X axis is different among the two treatment groups in the top row of the figure and the next row. Please clarify and re-label. Also, the number of animals represented is unclear and the legend or figure itself should identify this.

6) Line 470 states “prolonged treatment favored the development of a humoral response to SIV antigens...” I don’t think the data suggest that as 24 months of therapy nearly extinguishes detectable antibody. Please clarify.

7) This study did not assess antibody effector functions which is far more useful than the reported magnitudes of responses. Is this data available? If so, it should be included.

8) Line 602 relates the analysis in fig7D but it is unclear whether the data reflect the full 11 or 12 animals in each group or a subset. Please clarify.

9) Line 644-651 of the discussion states that very early treatment prevents antiviral responses to develop. However, there is published evidence of CD8 response arising around peak viremia and these are associated with epitope specific mutation and changes in dominant viral quasispecies. Pathogen specific antibody develops at peak viremia but full antibody immune response takes a longer period of time. I don’t disagree with your concept here but the statement as written may need qualification and couched more as a hypothesis. In particular, the number of persons with very early treatment who have undergone ATI and quickly rebound support your statement but some proportion do become PTC and we cannot say this is lower than might be observed if treatment was delayed to 4 weeks as I know of no experimental data corresponding to this approach in humans or animals. You did not have such an arm in this study. It might have been useful as it is standard to start ART as soon as one finds someone is HIV infected even if they are antibody negative, ie very early acute infection. I do not think it is necessary to do an additional arm of NHP treated at day 7 now but if you do replicate this experiment in future it might be considered.

10) Line 669: Although the paper is generally well written and clear, there are some issues with syntax, word choice and missing conjunctions etc that should be addressed. This is an example I think...“groups of macaques only differentiated by the delay” might be better stated as “groups of macaques only distinguished by”...or “macaques differing only by...”

11) I think this line starting at 709 needs clarification: “In the natural host, where the epithelial barrier is preserved as well, higher IgA levels than in macaques have been reported”. I think the authors are referring here to the responses in African Green monkeys which tolerate high levels of viremia without gut or systemic inflammatory responses and the paper referenced suggests a hypothesis for this observation. However, that is not the case in PTC who have low to undetectable levels of viremia. If the authors are trying to connect the observation in AGM versus macaques and PCT the introduction and discussion need to better explain this.

12) I would note that the antibody data, both IgG and IgA lack any functional assessment and I am wondering if this information could be reduced somewhat and perhaps placed in the supplement as there is not a clear value in the absence of effector data. The relationship between antigen exposure and antibody responses is well known.

13) Figure 6 shows the evolution of T cell subsets in acute and post ATI separately for the W4 group (figure 6A) and W24 (6B) in blood and similarly in 6C and 6D for LN. Lines 552-560 discuss the data in these figures with an apparent focus on comparing primary infection T cell subsets to post ATI but the key to this study is comparing the W4 group to W24 and patterns seem similar. There is nothing in the

discussion to frame the significance of the data shown in figure 6A-6D. I would delete or move to the supplement and shorten the associated narrative or better define the value of these observations.

Reviewer #3 (Remarks to the Author):

Passaes and colleagues present an SIV/macaque study comparing the relative effectiveness of combination anti-retroviral therapy to suppress SIV replication, impacting on viral reservoirs in deep lymphoid tissue and analyses of compartmentalisation of immune responses associated with suppression or control of virus. Rather than initiate therapy during the first few days post infection (eclipse phase) they compare treatment commencing at 4 and 24 weeks post exposure to a stock of SIVmac251. The authors employ Mauritian-derived cynomolgus macaques, which the group has a lot of previous experience with, where well characterised immunogenetic genotyping is a feature of this model. Virus dynamics are compared by conventional SIV RNA monitoring in plasma (viraemia) and cell-associated virus levels by SIV DNA PCR. Humoral and cellular responses are compared and a key feature of SIV/macaque studies exploited by virtue of the ability to sample multiple lymphoid tissues following animal sacrifice. The study is reasonably comprehensive in nature and well reported representing an important study which addresses a key issue in HIV-treatment of when to commence anti-retroviral therapy, with the delay in treatment having to balance a potentially enhanced immune response which may develop with reducing the seeding of virus to lymphoid tissue and reservoir establishment, which has represented received wisdom on this topic (hit hard, hit early).

There are a number of areas the authors should clarify.

SIVmac251 stock used should be described in more detail, including more details of origin and propagation and ideally any next generation sequencing data to confirm stock integrity, viral sub-populations etc. This would be helpful in making comparisons with other SIVmac251 challenges that have been performed, frequently using the rhesus model.

There is clearly some rebound in the ART-W4 group in Fig 1A and the authors could comment more on this, which may reflect the seeding of virus into reservoirs in lymphoid tissue (or at least that is the inference). It is not entirely clear at first glance what the long-term steady state viraemia of the SIVmac251 stock is. This data is included in the manuscript FigS1C (naïve, untreated controls) but would be better included perhaps as an overlay or summary or as a comparator figure as a part of Figure 1A. It looks like the steady state viraemia of untreated macaques is in the region of 10^5 log₁₀ RNA with a peak at days p.i. which is typical in cynos, although CCA100 and CCC039 suggest outliers above and below this of approx. 1 log. This suggests some differences in natural control of the SIV251 stock used and possible immunogenetic factors at play. CBK061 and CDJ052 look like they displays significant natural virus control. As the authors genotyped their Mauritian animals, more explanation is required as to what impact such host genetics have on naïve infection dynamics and in influencing the described

treatment responses; in treatment groups there are also differences in immediate post peak infection control of viraemia and rebound of viraemia eg BA912K, BB425F.

It is perhaps surprising SIV-DNA levels did not differ significantly between W4 and W24 groups as it would seem intuitive that rebound of virus from viral reservoirs would be impacted by differences in treatment regimes where this would typically be demonstrated in the tissues selected for analysis. While SIV DNA levels, although a cruder measure of virus activity, do tend to reflect broader measures of virus replication and expression of SIV DNA against CD4 cell counts is valid where the relationship between SIV RNA in plasma and SIV DNA levels seems to tally, an analysis of SIV RNA levels in tissues would have more informative and, in this reviewer's opinion, a better indication of relative virus turnover and persistence in lymphoid tissue. This would deliver a more accurate measure of virus activity/signals in lymphoid compartments and potentially better differentiate W4 and W24 treatment groups. This would inevitably provide greater insight into the most important reservoirs for immune surveillance activity to suppress or contain as active virus turnover (by virtue of vRNA in tissues) would represent an immune stimulus the authors are purporting would be beneficial. Inclusion of this type of data would improve the manuscript and its conclusions. Did the authors attempt to measure or have any data relating to virus RNA levels in tissues? If so this should be included.

Similarly, extending the analysis of virus stock composition, would be (deep) sequence analysis of the rebound virus (ideally RNA) correlating plasma RNA sequence with lymphoid compartments which would further inform which tissues are most relevant in the context of persistent reservoirs. The authors comment on passing in the discussion on frequency of impact genomes post treatment in other studies. Any evidence of this here?

Some of the humoral data is not particularly convincing, although well-presented, and tends to suggest responses reflect antigenic stimulation following a virus rebound event rather than have any cause/effect relationship with virus control. The speculation in the discussion around IgA levels could be scaled back accordingly.

Despite these reservations, the cellular and humoral data are well executed and presented although the lack of any significant data relating to innate immunity is a weakness of this study. The comparisons drawn with HIV controllers, with HIV-2 providing a helpful comparator, speaks in part to the notion that enhanced CD8+ T cell activity can be correlated with virus control/suppression although which one is the driver in reaching this point remains (and the authors note) unanswered. Suppression of SIV by ART leading to virus control has many parallels with natural control of HIV-2 which may have as much to do with viral factors and immunologic ones. The authors could expand this point given the similarities between SIV and HIV-2, both in macaque models and outcomes of HIV-2 in humans.

Line 489-90 English needs correcting.

ART initiation did not immediately impact(ed) the capacity...

Line 651 ..selected.. better than 'picked' ?

Line 669 'only differenced' is clumsy English.

Line 755 stating 'unpublished observations' should be removed.

Line 775 '..and eventually other infectious diseases' is perhaps over-reaching here and should omitted from the final sentence.

REVIEWER COMMENTS

Reviewer #1 (Remarks to the Author):

The study was very well designed – including maintaining animals on ART for two years – and performed. The manuscript is very well written and includes a large amount of work convincingly showing that early ART strongly promote durable PTC in SIV-infected cynomolgus macaques and that this control was associated with the generation/expansion of long-term memory CD8 T cells with enhanced antiviral activities. These findings are very important and significant.

We thank the reviewer for their positive assessment of our study

I do have few comments to further improve the quality of the manuscript, as listed below:

Main Comments:

1. Since control is associated with the generation/expansion of long-term memory CD8 T cells, it will be interesting to further characterize those cells, including for the expression of immune checkpoint inhibitors and other markers that have been associated with T cell function, differentiation status, and longevity (for example PD-1, Tox, TCF-1, CD39).

Following the reviewer's suggestion, we explored the memory phenotype of the SIV-specific CD8+ T cells in the animals in our study. Due to sample availability, we focused our analyses on cells isolated from the spleen of the animals at the end of the study. We found that memory (CD45RA^{neg}) CD8+ T cells responding to SIV peptides from the W4-treated macaques were distinguished by higher expression of CCR7 and lower expression of CD39 when compared to cells from the macaques in the W24 groups. We also found a trend to higher expression of the transcription factor TCF-1, which has been associated with stemness potential, in the cells from the W4-treated macaques.

Proportion of SIV-specific memory CD8+ T cells expressing PD-1, CD39, CCR7 or TCF1 in the spleen of W4- and W24-treated macaques at the end of the study

A higher expression of CCR7 and TCF1 on SIV-specific memory CD8+ T cells was even more evident in animals classified as PTC vs non-PTC. The cells from non-PTCs were in contrast characterized by higher levels of PD-1.

Proportion of SIV-specific memory CD8+ T cells expressing PD-1, CD39, CCR7 or TCF1 in the spleen of PTC and Non-PTC macaques at the end of the study.

Overall, these results support the presence of longer-live memory SIV-specific CD8+ T cell responses with stemness in the early treated animals, while a more differentiated, prone to exhaustion profile was present in the cells from the late treated animals. We have included these results that nicely complement our initial results in the revised version of the manuscript (Figure 7 and Figure S9).

2. The authors show a quite impressive difference among the two groups in the levels of CM CD8 T cells in the colon and speculate this can be due to different levels of gut integrity, inflammation, and microbial translocation among the early and late ART groups. This concept can be tested by measuring soluble markers of inflammation/gut integrity (LPS, LBP, sCD14, etc.) and/or levels of IL-17 producing CD4 T cells, that has been indicated as critical for gut integrity.

To address this point, we analyzed at different times the plasma levels of sCD14 and the frequency of circulating CD4+ T cells able to produce IL-17 after polyclonal stimulation. Although we observed longitudinal changes in these parameters (particularly, an increase of sCD14 levels and drop in the frequency of Th17 cells in most animals before treatment initiation), no differences were observed between the groups of animals depending on the delay to treatment initiation (or when classified as PTC or non-PTC). A trend for reduced Th17 cells was perhaps observed for the W24 group at the end of the study, but this was not sufficient to sustain the hypothesis concerning gut integrity.

Changes in the frequency of CD4+ T cells producing IL-17 in response to concanavalin A+IL2 (left) and of plasma sCD14 (right) at different follow-up times for W4- and W24- treated macaques, when compared to baseline.

We have preliminary data indicating the preservation of CX3CR1+ macrophages in the gut of PTC macaques, which may contribute to the maintenance of the intestinal epithelial barrier in these animals. We will explore further this question, but we decided to remove this speculation from the discussion in the revised manuscript.

3. The study very convincingly shows different level of IL-7R expression in SIV-specific CD8 T cells among the two groups. Are the levels of IL-7 different at ATI and post ATI?

Despite our best efforts we could not quantify IL-7 levels so far. There was a supply shortage for the ELISA kit that we had previously validated for this model (doi.org/10.1128/2FJVI.03723-14). We tried to quantify IL-7 by Luminex but IL-7 levels were not detectable with this technique. We decided not to delay further the submission of the revised version of the manuscript, but we will explore the changes in the levels of IL-7 and other relevant cytokines in future analyses.

Minor Comments:

1. Were the levels of the CD8 T cell differentiation subsets (SCM, CM, EM) different among the two groups at ATI?

No differences were observed between the groups at the time of treatment interruption. We have now clarified this in the text of the revised version of the manuscript (lines 596-601)

Frequency of CD8+ T cells with central memory (CM), transitional memory (TM), effector memory (EM) or effector phenotypes in W4- and W24-treated macaques at the time of ART interruption

2. The authors showed an interesting correlation between the expansion of CM CD8 T cells and the suppressive activity of CD8 T cells, both in blood and LN. Is there a correlation between the frequency of proliferating (Ki-67+) CM CD8 T cells and the level of CM CD8? What about the frequency of proliferating CM CD8 T cells and the suppressive activity?

No correlations were found between the frequency of proliferating Ki67+ CM CD8+ T cells and the frequency of CM CD8+ T cells in blood or PLN, but a correlation was found between the frequency of proliferating Ki67+ CM CD8+ T cells and the suppressive activity of CD8+ T cells in blood.

Spearman correlation between CD8+ T-cell-mediated SIV-suppressive activity AUC post-ATI and the frequency of Ki67+ CM CD8+ T cells 28 days post-ATI in blood (left) and 14 days post-ATI in PLNs (right).

Reviewer #2 (Remarks to the Author):

A critical insight here is that the field should look more formally at timing of interventions despite the ample evidence that treatment during AHI reduces the reservoir and preserves immune integrity. However critically, we need to better understand the viral and immune parameters associated with PCT. Here we have evidence of CD8 suppression of autologous CD4 cell SIV infection associating with the 4 week ART group showing a trend to higher CD8 suppressive activity prior to ATI which increases during the ATI. No other measured markers seem to correlate with viral control though IL 7 expression was higher in the W4 group and among all PCT animals compared to non PCT animals. Taken together the paper demonstrates a dramatic increase in PTC among the W4 group with no evidence of benefit among the W24 ART group compared to no ART groups and W4 ART favored establishment of memory like SIV specific CD8 cells with enhanced proliferative potential and survival capacity. These are important observations but it seems there is much left to analyze which may be far more informative by an in depth analysis of other reservoir measurements in different T cell compartments and additional functional assay work for innate responses and humoral effectors.

We thank the reviewer for their positive evaluation and suggestions. Indeed, the pVISCONTI study is a very large endeavor involving multiple groups with diverse expertise. The present manuscript represents the princeps article describing the outcome of the study and an initial assessment of virological and immunological markers. Additional sub-studies are ongoing focusing, among others, on viral reservoirs, innate immunity, inflammation and gut integrity, pharmacology, or mathematical modelling, to provide a through characterization of the dynamics of post-treatment control.

Comments:

1) Methods: the laudable and ethical conditions of NHP management in your study are more detailed in lines 96-117 than needed for the reader and can be shortened (not eliminated) with content shifted as appropriate to the supplement.

As suggested by the reviewer, we have shortened this section in the manuscript and placed the complete description into the reporting summary

2) Methods: This sentence starting on line 132 is a bit unclear: Most animals without post treatment control reached clinical scores indicating euthanasia before the pre-scheduled follow up period, but with at least 6 months of follow up after treatment interruption (Figure S1, Tables S1). Please clarify.

We have modified this part as follows (lines 135-138):

“The pre-scheduled end of the study was 48 weeks post-treatment interruption. However, most animals without post-treatment control progressed to clinical scores requiring euthanasia before this time. Nevertheless, all animals were monitored for at least 6 months after treatment interruption (Figure S1, Tables S1)”.

3) Methods: It is not state of the art to measure total HIV DNA as TILDA or IPDA measurements better reflect the rebounding, ie replication competent reservoir and given the tissue collections, determining the cell phenotype of reservoir cells might be feasible and very informative as well. Despite its limitations, the analysis of cell-associated total viral DNA is the marker/estimate of the viral reservoir, with high reproducibility, that has provided the wider information of the distribution of infected cells in the body, at all stages of HIV disease and during therapy (<https://doi.org/10.1128/cmr.00015-16>), and the only one, to our knowledge, that has shown some predictive value of post-treatment control in clinical settings (<https://doi.org/10.7554/eLife.03821>; <https://doi.org/10.1097/QAD.0000000000000734>; <https://doi.org/10.1097/qad.0000000000000987>). Our results indicate that there were not major differences in the frequency of infected cells between the early- and the late-treated groups that may explain the different outcome of the animals upon ART interruption.

The reviewer is right to point out that there are new approaches that allow a deeper characterization of the quality/composition of the viral reservoirs. Additional virological analyses, including IPDA and near-full-length genome sequencing, are undergoing in multiple tissues to analyze the phylodynamic of infection and characterize the viral reservoir in detail. However, these techniques have needed adaptation to our model, the results are not yet available and will be the subject of another manuscript. To provide a more complete characterization of the virological situation of the animals at the time of ART interruption we have analyzed and added to the manuscript (Figure S8 and lines 381-383) data showing that no differences were found either in the levels of cell-associated viral RNA in PBMCs or lymph node cells at that moment.

Comparison of cell-associated SIV RNA in blood and PLN of W4- and W24-treated CyMs at the at the time of treatment interruption.

4) Methods: the endpoint definition is noted in the results but indicates PTC required VL <400 copies per mL but not whether this needed to be a durable result over several measurements or just the last study measurement. This should be explicitly stated in the methods as the primary endpoint and key definitions like “rebound” should be included as well.

Animals were defined as PTC when they reached at least one VL <400 RNA copies/ml following the viral rebound upon ART interruption (or if never rebounded). In practice, all PTCs had at least 2 VL<400 copies RNA/ml after controlling viral rebound and all, except BA736J (last VL=514 RNA/ml), maintained their VL<400 copies until the end of the study. We have clarified this in the new version of the manuscript (lines 131-134).

5) Line 340 references the figure S1C...a critical figure depicting VL over time in the 7 treatment and control experiments (2 groups of W4, 2 groups of W24 and 3 groups with no ART). I am trying to figure out the labels in the figure for W4 and W24 week treatment groups as the X axis is different among the two treatment groups in the top row of the figure and the next row. Please clarify and re-label. Also, the number of animals represented is unclear and the legend or figure itself should identify this.

In the new version of the manuscript, we have separated the study design and the viral loads in the different groups in two figures (Figure S1 and Figure S7). We also realized that the panels for the pVISCANTI-2 groups were mislabeled: top panels correspond to ART W4 animals and bottom panels to ART W24 animals, which explains the different axis (ART initiation at different time points). We thank the reviewer for pointing out this discrepancy.

6) Line 470 states “prolonged treatment favored the development of a humoral response to SIV antigens...” I don’t think the data suggest that as 24 months of therapy nearly extinguishes detectable antibody. Please clarify.

As the reviewer indicated, the time on ART led to a drop in the anti-SIVgp140 antibody levels to baseline levels in most animals. However, the humoral response seemed to increase faster upon viral rebound than what observed during primary infection (see below), suggesting a reactivation of SIVgp140-specific memory B cells that persisted during the 2 years of ART, and the subsequent, quicker, antibody production upon ATI. We now indicate and discuss these differences in the IgG and IgA titers in the revised manuscript (lines 508-510 and figure S10).

Evolution of anti-gp140 IgG and IgA titers from baseline to 28 days p.i. and from time of ART interruption to 28 days post-ATI in W4- and W24-treated macaques. Repeated measures 2-way ANOVA.

7) This study did not assess antibody effector functions which is far more useful than the reported magnitudes of responses. Is this data available? If so, it should be included.

The antibody effector functions as well as the SIV neutralizing capacities of purified serum IgGs were not analyzed here but will be investigated in the future. Indeed, we have recently reported that some human PTCs developed antibodies with cross-neutralizing and ADCC activities (10.1038/s41467-022-29511-1 & 10.1016/j.chom.2023.06.006). We have indicated this limitation in the discussion of our manuscript (lines 510-512 and 759-765).

8) Line 602 relates the analysis in fig7D but it is unclear whether the data reflect the full 11 or 12 animals in each group or a subset. Please clarify.

We performed those analyses with cells from the animals in the pVISCONTI-1 groups, which samples were destined to deeper immunological characterization. We have clarified this in the legend of the figure.

9) Line 644-651 of the discussion states that very early treatment prevents antiviral responses to develop. However, there is published evidence of CD8 response arising around peak viremia and these are associated with epitope specific mutation and changes in dominant viral quasispecies. Pathogen specific antibody develops at peak viremia but full antibody immune response takes a longer period of time. I don't disagree with your concept here but the statement as written may need qualification and couched more as a hypothesis. In particular, the number of persons with very early treatment who have undergone ATI and quickly rebound support your statement but some proportion do become PTC and we cannot say this is lower than might be observed if treatment was delayed to 4 weeks as I know of no experimental data corresponding to this approach in humans or animals. You did not have such an arm in this study. It might have been useful as it is standard to start ART as soon as one finds someone is HIV infected even if they are antibody negative, ie very

early acute infection. I do not think it is necessary to do an additional arm of NHP treated at day 7 now but if you do replicate this experiment in future it might be considered.

We thank the reviewer for their comment. We have clarified in the text that when we refer to extremely early treatment preventing the development of antiviral responses, we were pointing to treatment initiation before reaching the peak of viremia (lines 694, 696-697). We have also mitigated this statement. Most PTCs reported to date started treatment in Fiebig III-V stages (and this is what guided our experimental design), whereas no PTCs have been reported to our knowledge among people who started ART in Fiebig I-II stages, although the number of such people is very limited.

We initially thought about including animals treated within the first week post- infection, but previous results by others and us in different macaque models (albeit with shorter treatment duration) showed that post-treatment control was not reached when ART was initiated during this period (<https://doi.org/10.1186/1742-4690-7-78>; <https://doi.org/10.1038/nature13594>). Therefore, we decided to focalize our resources (number of animals and replication of the study) in testing the hypothesis of treatment initiation during the period more often observed in human post-treatment controllers from the VISCONTI cohort vs treatment during chronic infection. We agree that the clear results obtained in the present study encourage further experiments to determine more precisely the window of opportunity for treatment initiation favoring post-treatment control.

10) Line 669: Although the paper is generally well written and clear, there are some issues with syntax, word choice and missing conjunctions etc that should be addressed. This is an example I think...“groups of macaques only differenced by the delay” might be better stated as “groups of macaques only distinguished by”...or “macaques differing only by...”

The new version of the manuscript has been reviewed by professional English editors

11) I think this line starting at 709 needs clarification: “In the natural host, where the epithelial barrier is preserved as well, higher IgA levels than in macaques have been reported”. I think the authors are referring here to the responses in African Green monkeys which tolerate high levels of viremia without gut or systemic inflammatory responses and the paper referenced suggests a hypothesis for this observation. However, that is not the case in PTC who have low to undetectable levels of viremia. If the authors are trying to connect the observation in AGM versus macaques and PCT the introduction and discussion need to better explain this.

We agree with the reviewer that this mention can lead to confusion between the models and has been removed in the new version of the manuscript

12) I would note that the antibody data, both IgG and IgA lack any functional assessment and I am wondering if this information could be reduced somewhat and perhaps placed in the supplement as there is not a clear value in the absence of effector data. The relationship between antigen exposure and antibody responses is well known.

We have reduced the description and discussion of this data and acknowledge the limitation in the absence of functional testing and results, but we believe that the immunomonitoring data on the specific immunoglobulin titers contributes to expose the differences in the dynamics of antigen exposure and immune response in the early- vs late- treated animals

13) Figure 6 shows the evolution of T cell subsets in acute and post ATI separately for the W4 group (figure 6A) and W24 (6B) in blood and similarly in 6C and 6D for LN. Lines 552-560 discuss the data in these figures with an apparent focus on comparing primary infection T cell subsets to post ATI but

the key to this study is comparing the W4 group to W24 and patterns seem similar. There is nothing in the discussion to frame the significance of the data shown in figure 6A-6D. I would delete or move to the supplement and shorten the associated narrative or better define the value of these observations.

Since we observed a dramatic increase in the capacity of CD8+ T cells to suppress infection during ART interruption when compared to primary infection (when this activity was not present), our objective here was to expose differences in the expansion of CD8+ T cell subsets during these two periods that could be relevant to better understand this enhancement. Our results showed that CD8+ T cells expanding in response to primary infection mainly had an effector memory or effector phenotype, cells expanding after treatment interruption in the early treated group had a central memory phenotype. We believe that these results are important as they suggest that early and prolonged treatment favored the establishment of central memory CD8+ T cells that were able to better respond to viral rebound. Some changes were also observed in the profile of the cells expanding during primary infection and after ART interruption in the W24 group, however these differences were less dramatic than for the W4 animals, as late-treated animals still showed a limited expansion of cells with central memory phenotype and a clear expansion of cells with effector phenotype after ART interruption (see below). We have tried to better describe these observations and our interpretation in the revised version of the manuscript (lines 596-601 and 787-793).

Direct comparison of the % of CD8+ T cells with central memory or effector phenotype in blood and PLN of W4- and W24-treated macaques at different times following ART interruption

Reviewer #3 (Remarks to the Author):

The study is reasonably comprehensive in nature and well reported representing an important study which addresses a key issue in HIV-treatment of when to commence anti-retroviral therapy, with the delay in treatment having to balance a potentially enhanced immune response which may develop with reducing the seeding of virus to lymphoid tissue and reservoir establishment, which has represented received wisdom on this topic (hit hard, hit early). There are a number of areas the authors should clarify.

We thank the reviewer for their positive consideration of our study.

SIVmac251 stock used should be described in more detail, including more details of origin and propagation and ideally any next generation sequencing data to confirm stock integrity, viral sub-populations etc. This would be helpful in making comparisons with other SIVmac251 challenges that have been performed, frequently using the rhesus model.

We have now provided more information on the origin and propagation of the virus stock inoculum.

Cell-free virus stock of pathogenic SIVmac251 was kindly provided by A. M Aubertin (Université Louis Pasteur, Strasbourg, France). The virions were obtained from the cell-free supernatant of infected rhesus peripheral blood mononuclear cells (PBMC). Cells were infected in vitro with a culture supernatant obtained from a co-culture of rhesus PBMC and a spleen homogenate from a rhesus macaque infected with SIVmac251 (provided by R. C. Desrosiers, New England Regional Primate Center, Southborough, Mass.).

We are performing near-full-length genome sequencing to analyze the phylodynamic of infection in the pVISCNTI study (including the composition of the viral stock), but these results are not available yet.

There is clearly some rebound in the ART-W4 group in Fig 1A and the authors could comment more on this, which may reflect the seeding of virus into reservoirs in lymphoid tissue (or at least that is the inference). It is not entirely clear at first glance what the long-term steady state viraemia of the SIVmac251 stock is. This data is included in the manuscript FigS1C (naïve, untreated controls) but would be better included perhaps as an overlay or summary or as a comparator figure as a part of Figure 1A. It looks like the steady state viraemia of untreated macaques is in the region of 10^5 log₁₀ RNA with a peak at days p.i. which is typical in cynos, although CCA100 and CCC039 suggest outliers above and below this of approx. 1 log. This suggests some differences in natural control of the SIV251 stock used and possible immunogenetic factors at play. CBK061 and CDJ052 look like they displays significant natural virus control. As the authors genotyped their Mauritian animals, more explanation is required as to what impact such host genetics have on naïve infection dynamics and in influencing the described treatment responses; in treatment groups there are also differences in immediate post peak infection control of viraemia and rebound of viraemia eg BA912K, BB425F.

We thank the reviewer for their suggestion. We have now overlaid in figure 1A for reference the IQR of the viral loads during the first 250 days post infection for the animals naïve of antiretroviral treatment. We excluded in this study all animal carrying the M6 MHC haplotype than other and we have shown to strongly favor natural SIV control in this model (e.g. 10.1016/j.celrep.2020.108174). However, we cannot exclude that not yet identified immunogenetic factors also influence infection. To minimize these potential effects, we tried to match as best as possible the groups of study based

on their MHC haplotypes. We directly mention now the potential role of immunogenetic factors when discussing the outcome of the animals (lines 406-411).

Plasma viral load kinetics prior to ART initiation, during ART; and post-ATI in W4-treated (blue) and W24-treated (orange) CyMs. Plasma viral load during the initial six months following infection in animals not receiving ART shown as reference (grey). Medians and IQR are shown.

It is perhaps surprising SIV-DNA levels did not differ significantly between W4 and W24 groups as it would seem intuitive that rebound of virus from viral reservoirs would be impacted by differences in treatment regimes where this would typically be demonstrated in the tissues selected for analysis. While SIV DNA levels, although a cruder measure of virus activity, do tend to reflect broader measures of virus replication and expression of SIV DNA against CD4 cell counts is valid where the relationship between SIV RNA in plasma and SIV DNA levels seems to tally, an analysis of SIV RNA levels in tissues would have more informative and, in this reviewer's opinion, a better indication of relative virus turnover and persistence in lymphoid tissue. This would deliver a more accurate measure of virus activity/signals in lymphoid compartments and potentially better differentiate W4 and W24 treatment groups. This would inevitably provide greater insight into the most important reservoirs for immune surveillance activity to suppress or contain as active virus turnover (by virtue of vRNA in tissues) would represent an immune stimulus the authors are purporting would be beneficial. Inclusion of this type of data would improve the manuscript and its conclusions. Did the authors attempt to measure or have any data relating to virus RNA levels in tissues? If so this should be included.

Following the reviewer's suggestion, to complete our observation not showing differences between W4 and W24 groups in SIV DNA levels after two years of treatment, we analyzed cell-associated viral RNA levels in PBMCs and lymph node cells at the time of ART interruption. These analyses did not show either any significant difference between the groups of animals. We have included this information in the revised version of the manuscript (figure S8, lines 381-383).

Comparison of cell-associated SIV RNA in blood and PLN of W4- and W24-treated CyMs at the at the time of treatment interruption.

Similarly, extending the analysis of virus stock composition, would be (deep) sequence analysis of the rebound virus (ideally RNA) correlating plasma RNA sequence with lymphoid compartments which would further inform which tissues are most relevant in the context of persistent reservoirs. The authors comment on passing in the discussion on frequency of impact genomes post treatment in other studies. Any evidence of this here?

As mentioned in our response to Reviewer 2, additional virological analyses, including IPDA and high-throughput sequencing, are undergoing in multiple tissues to analyze the phylodynamic of infection and characterize in detail the viral reservoir and the rebounding virus. However, the results are not yet available and will be the subject of another manuscript. For disclosure, we do not currently have any results revealing differences in the viral reservoir before or during ART that could explain the different outcome after ART interruption between the W4-treated and W24-treated animals in the study.

Some of the humoral data is not particularly convincing, although well-presented, and tends to suggest responses reflect antigenic stimulation following a virus rebound event rather than have any cause/effect relationship with virus control. The speculation in the discussion around IgA levels could be scaled back accordingly.

We have reduced the description and discussion of this data and acknowledge the limitation in the absence of functional data (lines 510-512 and 759-765).

Despite these reservations, the cellular and humoral data are well executed and presented although the lack of any significant data relating to innate immunity is a weakness of this study.

We appreciate the importance that innate immunity may have in the context of post-treatment control of infection. The present manuscript describes the outcome of the study and an initial assessment of virological and immunological markers. Additional specific sub-studies are ongoing focusing, among other, on innate immunity, viral reservoirs (as mentioned), inflammation, pharmacology, or mathematical modelling, to provide a through characterization of the dynamics of post-treatment control.

The comparisons drawn with HIV controllers, with HIV-2 providing a helpful comparator, speaks in part to the notion that enhanced CD8+ T cell activity can be correlated with virus control/suppression although which one is the driver in reaching this point remains (and the authors note) unanswered. Suppression of SIV by ART leading to virus control has many parallels with natural control of HIV-2

which may have as much to do with viral factors and immunologic ones. The authors could expand this point given the similarities between SIV and HIV-2, both in macaque models and outcomes of HIV-2 in humans.

We thank the reviewer for the suggestion, we discuss now how viral factors might contribute to the development of CD8+ T cells with enhanced activity (lines 777-779)

English needs correcting.

Line 489-90 ART initiation did not immediately impact(ed) the capacity...

Line 651 ..selected.. better than 'picked' ?

Line 669 'only differenced' is clumsy English.

We apologize for these mistakes; all have been corrected. The new version of the manuscript has been revised by a professional English editor.

Line 755 stating 'unpublished observations' should be removed.

This mention was removed

Line 775 ..and eventually other infectious diseases' is perhaps over-reaching here and should omitted from the final sentence.

We removed this comment in the revised version of the manuscript

REVIEWERS' COMMENTS

Reviewer #1 (Remarks to the Author):

The authors addressed all the main comments from this and the other reviewers.

Reviewer #2 (Remarks to the Author):

The authors have considered comments of the reviewers and provided very reasonable responses and corresponding modifications to the manuscript. However, I do wonder if there should be some caveat related to the following lines in the discussion:

813 In addition, in our model, all SIV PTCs except one experienced viral rebound after ATI. However, many PTCs in human studies appear to consistently maintain undetectable viral loads after ART interruption 13. Thus, SIV PTC may better recapitulate the cellular and humoral immune response during active control of infection after ART interruption, which is also often observed in HIV PTCs 6,11,13,59,73.

I am not sure I follow the statement: "Thus, SIV PTC may better recapitulate the immune response in HIV PTCs". It would seem to me that there remains the possibility that there are crucial determinants of PCT in humans to achieve undetectable levels of viremia which is without question the clinically meaningful goal. I might suggest modifying this section accordingly.

Reviewer #3 (Remarks to the Author):

The authors present a revised and improved version of their manuscript which has satisfactorily addressed my original concerns. I have no further comments the authors need to act on.

REVIEWER COMMENTS

Reviewer #1 (Remarks to the Author):

The authors addressed all the main comments from this and the other reviewers.

We thank the reviewer for their useful comments during the revision process

Reviewer #2 (Remarks to the Author):

The authors have considered comments of the reviewers and provided very reasonable responses and corresponding modifications to the manuscript. However, I do wonder if there should be some caveat related to the following lines in the discussion:

813 In addition, in our model, all SIV PTCs except one experienced viral rebound after ATI. However, many PTCs in human studies appear to consistently maintain undetectable viral loads after ART interruption¹³. Thus, SIV PTC may better recapitulate the cellular and humoral immune response during active control of infection after ART interruption, which is also often observed in HIV PTCs^{6,11,13,59,73}.

I am not sure I follow the statement: “Thus, SIV PTC may better recapitulate the immune response in HIV PTCs”. It would seem to me that there remains the possibility that there are crucial determinants of PCT in humans to achieve undetectable levels of viremia which is without question the clinically meaningful goal. I might suggest modifying this section accordingly.

The text was indeed confusing. Observational studies suggest that some human PTCs maintain strictly undetectable control of viremia after treatment interruption while others are able to recontrol viral rebound or episodes of viremia after treatment interruption. We wanted to indicate that our model appears best fitted to study the latter. We acknowledge that there may be determinants in human PTCs that are not present in the macaques.

We have modified the text as follows:

Many human PTCs appear to consistently maintain undetectable viral loads after ART interruption¹³, while others experience transient episodes of viremia^{6,11,13,51,65}. In our model, all SIV PTCs except one experienced viral rebound after ATI and may better recapitulate an active control of infection after ART interruption. Some determinants of viral control may differ between macaques and human PTCs. For instance, some immunogenetic traits that have been proposed to favor post-treatment HIV control in PWH⁶⁶ may be absent in CyMs⁶⁷.

We thank the reviewer for their useful comments during the revision process.

Reviewer #3 (Remarks to the Author):

The authors present a revised and improved version of their manuscript which has satisfactorily addressed my original concerns. I have no further comments the authors need to act on.

We thank the reviewer for their useful comments during the revision process